# Gon4l regulates notochord boundary formation and cell polarity underlying axis extension by repressing adhesion genes

Margot L.K. Williams[1], Atsushi Sawada[1,2], Terin Budine[1], Chunyue Yin[2,3], Paul Gontarz[1] & Lilianna Solnica-Krezel[1,2]

Anteroposterior (AP) axis extension during gastrulation requires embryonic patterning and morphogenesis to be spatiotemporally coordinated, but the underlying genetic mechanisms remain poorly understood. Here we define a role for the conserved chromatin factor Gon4l, encoded by *ugly duckling* (*udu*), in coordinating tissue patterning and axis extension during zebrafish gastrulation through direct positive and negative regulation of gene expression. Although identified as a recessive enhancer of impaired axis extension in planar cell polarity (PCP) mutants, *udu* functions in a genetically independent, partially overlapping fashion with PCP signaling to regulate mediolateral cell polarity underlying axis extension in part by promoting notochord boundary formation. Gon4l limits expression of the cell–cell and cell–matrix adhesion molecules EpCAM and Integrinα3b, excesses of which perturb the notochord boundary via tension-dependent and -independent mechanisms, respectively. By promoting formation of this AP-aligned boundary and associated cell polarity, Gon4l cooperates with PCP signaling to coordinate morphogenesis along the AP embryonic axis.

[1] Department of Developmental Biology, Washington University School of Medicine, Saint Louis, MO 63110, USA. [2] Department of Biological Sciences, Vanderbilt University, Nashville, TN 37235, USA. [3] Present address: Division of Pediatric Gastroenterology, Hepatology, and Nutrition, Cincinnati Children's Hospital, Cincinnati, OH 45229, USA. Correspondence and requests for materials should be addressed to L.S.-K. (email: solnical@wustl.edu)

Gastrulation is a critical period of animal development during which the three primordial germ layers—ectoderm, mesoderm, and endoderm—are specified, patterned, and shaped into a rudimentary body plan. During vertebrate gastrulation, an elongated anteroposterior (AP) axis emerges as the result of convergence and extension (C&E), a conserved set of gastrulation movements characterized by the concomitant AP elongation and mediolateral (ML) narrowing of the germ layers[1,2]. C&E is accomplished by a combination of polarized cell behaviors, including directed migration and ML intercalation behavior (MIB)[3–5]. During MIB, cells elongate and align their bodies and protrusions in the ML dimension and intercalate preferentially between their anterior and posterior neighbors[5]. This polarization of cell behaviors with respect to the AP axis is regulated by planar cell polarity (PCP) and other signaling pathways[6–10]. Because these pathways are essential for MIB and C&E but do not affect cell fates[7,10,11], other mechanisms must spatiotemporally coordinate morphogenesis with embryonic patterning to ensure normal development. BMP, for example, coordinates dorsal–ventral axis patterning with morphogenetic movements by limiting expression of PCP signaling components and C&E to the embryo's dorsal side[12]. In general, though, molecular mechanisms that coordinate gastrulation cell behaviors with axial patterning are poorly understood, and remain one of the key outstanding questions in developmental biology.

Epigenetic regulators offer a potential mechanism by which broad networks of embryonic patterning and morphogenesis genes can be co-regulated. Epigenetic modifiers form protein complexes with chromatin factors that are thought to regulate their binding at specific genomic regions in context-specific ways[13]. The identities, functions, and specificities of chromatin factors with roles during embryogenesis are only now being elucidated, and some have described roles in cell fate specification and embryonic patterning[14]. One such chromatin factor is Gon4l, whose homologs have conserved roles in cell cycle regulation and/or embryonic patterning in plants, worms, flies, mice, and fish[15–19]. However, the contribution of Gon4l or any other chromatin factor to morphogenesis is particularly poorly understood.

Here, we demonstrate a role for zebrafish Gon4l, encoded by ugly duckling (udu), as a regulator of embryonic axis extension during gastrulation. udu was identified in a forward genetic screen for enhancers of short axis phenotypes in PCP mutants, but we find it functions in parallel to PCP signaling. Instead, complete maternal and zygotic udu (MZudu) deficiency produces a distinct set of morphogenetic and cell polarity phenotypes that implicate the notochord boundary in ML cell polarity and cell intercalation during C&E. Extension defects in MZudu mutants are remarkably specific, as internalization, epiboly, and convergence gastrulation movements occur normally. Gene expression profiling reveals that Gon4l regulates expression of a large portion of the zebrafish genome, including genes with roles in housekeeping, patterning, and morphogenesis. Furthermore, Gon4l-associated genomic loci are identified by DNA adenine methyltransferase (Dam) identification[20,21] paired with high-throughput sequencing (DamID-seq), revealing both positive and negative regulation of putative direct targets by Gon4l. Mechanistically, we find that increased expression of epcam and itga3b, direct targets of Gon4l-dependent repression during gastrulation, each contribute to notochord boundary defects in MZudu mutants via a distinct molecular mechanism. This report thereby defines a critical role for a chromatin factor in the regulation of gastrulation cell behaviors in vertebrate embryos: by ensuring proper formation of the AP-aligned notochord boundary and associated ML cell polarity, Gon4l cooperates with PCP signaling to coordinate morphogenesis that extends the AP embryonic axis.

## Results

**Gon4l regulates axis extension during zebrafish gastrulation.** To identify previously unknown regulators of C&E, we performed a three generation synthetic mutant screen[22,23] using zebrafish carrying a hypomorphic allele of the PCP gene knypek (kny)/glypican 4, kny[m8188]. F0 wild-type (WT) males were mutagenized with N-ethyl-N-nitrosourea (ENU) and outcrossed to WT females. The resulting F1 fish were outcrossed to kny[m818/818] males (rescued by kny RNA injection) to generate F2 families whose F3 offspring were screened at 12 and 24 h post-fertilization (hpf) for short axis phenotypes (Fig. 1a). Screening nearly 100 F2 families yielded eight recessive mutations that enhanced axis extension defects in kny[m818/818] embryos (Fig. 1b–e). vu68 was found to be a L227P kny allele, and vu64 was a Y219* nonsense allele of the core PCP gene trilobite(tri)/vangl2, mutations in which exacerbate kny mutant phenotypes[24], demonstrating effectiveness of our screening strategy. We focused on vu66/vu66 mutants, which displayed a pleiotropic phenotype at 24 hpf, including shortened AP axes, reduced tail fins, and heart edema (Fig. 1d, Supplementary Fig.1b).

Employing the simple sequence repeat mapping strategy[25], we mapped the vu66 mutation to a small region on Chromosome 16 that contains the udu gene (Fig. 1f). udu encodes a conserved chromatin factor homologous to gon-4 in Caenorhabditis elegans[16] and the closely related Gon4l in mammals[18,19]. The previously described udu mutant phenotypes resemble those of homozygous vu66 embryos, including a shorter body axis and abnormal tail fins[18,26], making udu an excellent candidate for this PCP enhancer. Sequencing cDNA of the udu coding region from 24 hpf vu66/vu66 embryos revealed a T to A transversion at position 2261 predicted to change 753Y to a premature STOP codon (Fig. 1g). Furthermore, vu66 failed to complement a known udu[sq1] allele[18] (Supplementary Fig. 1c). Together, these data establish vu66 as an udu allele and identify it as a recessive enhancer of axis extension defects in kny PCP mutant gastrulae.

**Complete loss of udu function impairs axial extension.** To assess the full role of Gon4l during early development, we eliminated maternal expression of udu[18] using germline replacement[27]. The resulting WT females carrying udu[vu66/vu66] (udu−/−) germline were crossed to udu[vu66/vu66] germline males to produce 100% embryos lacking both maternal (M) and zygotic (Z) udu function, hereafter referred to as MZudu mutants. These mutants appeared outwardly to develop normally until mid-gastrula stages (Fig. 1h, i), but exhibited clear abnormalities at the onset of segmentation (Fig. 1j, k). MZudu−/− gastrulae specified the three germ layers (Fig. 1l–m, Supplementary Fig. 2), formed an embryonic shield marked by gsc expression (Supplementary Fig. 2d), and completed epiboly on schedule (Fig. 1h–i), but somites were largely absent in mutants (Fig. 1j, k, n, o). Although myoD expression was observed within adaxial cells by whole mount in situ hybridization (WISH), its expression was not detected in nascent somites (Fig. 1o), similar to reported descriptions of Zudu mutants[28]. Formation of adaxial cells is consistent with normal expression of their inducer shh in the axial mesoderm[29] (Supplementary Fig. 2h), and ntla/brachyury expression in the axial mesoderm was also largely intact (Supplementary Fig. 2j). Importantly, MZudu mutants were markedly shorter than age-matched WT controls throughout segmentation (Fig. 1j, k) and at 24 hpf (Supplementary Fig.3e, h). Although increased cell death was observed in MZudu mutants (as in Zudu mutants[28]), inhibiting apoptosis via injection of RNA encoding

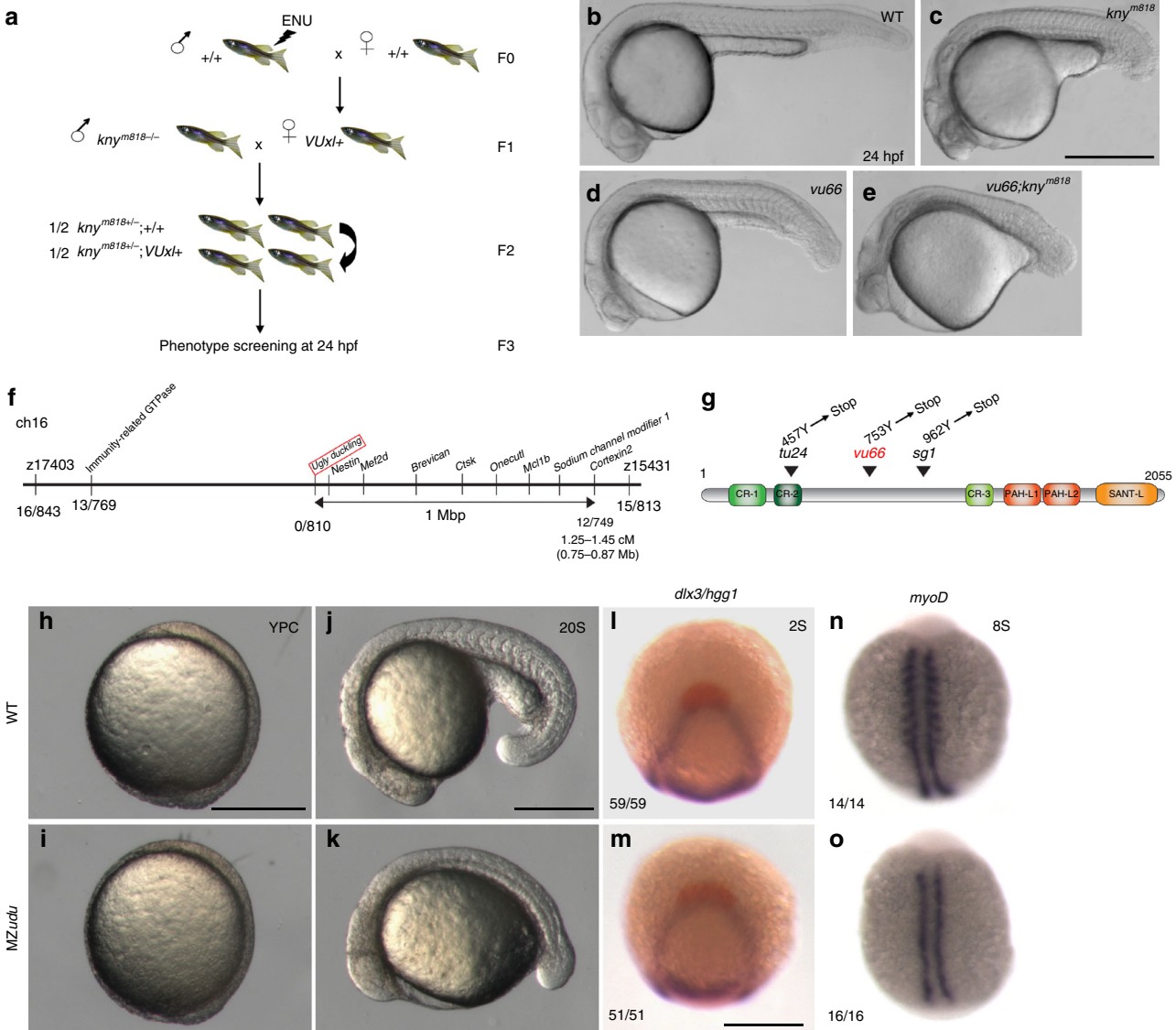

**Fig. 1** A forward genetic screen identifies *ugly duckling(udu)/gon4l* as a regulator of axis extension in zebrafish embryos. **a** Schematic of a synthetic screen to identify enhancers of the short axis phenotype in *kny^m818/m818* zebrafish mutants. **b–e** Phenotypes at 24 hpf: wild type (WT) (**b**), *kny^m818/m818* (**c**), *vu66/ vu66* (**d**), *kny^m818/m818*; *vu66/vu66* compound mutants (**e**). Images are representative of phenotypes observed at Mendelian ratios in multiple independent clutches. **f** Diagram of mapping *vu66* mutation to Chromosome 16. Bold numbers below specify the number of recombination events between *vu66* and the indicated loci. **g** Diagram of the Gon4l protein encoded by the *udu* locus. Arrowheads indicate residues mutated in *vu66* and other described *udu* alleles. **h–k** Live WT (**h, j**) and maternal zygotic (MZ)*udu* (**i, k**) embryos at yolk plug closure (YPC) (**h–i**) and 20 somite stage (**j–k**). 100 percent of MZ*udu* mutants from more than 15 germline-replaced females exhibited the pictured phenotypes. **l–m** Whole mount in situ hybridization (WISH) for *dlx3* (purple) and *hgg1* (red) in WT (**l**) and MZ*udu−/−* (**m**) embryos at two-somite stage. **n–o** WISH for *myoD* in WT (**n**) and MZ*udu−/−* (**o**) embryos at eight-somite stage. Anterior is to the left in **b–e, j–k**; anterior is up in **h–i, l–o**. Fractions indicate the number of embryos with the pictured phenotype over the number of embryos examined. Scale bar is 500 μm in **b–e** and 300 μm in **h–o**

the anti-apoptotic mitochondrial protein Bcl-xL[30] did not suppress their short axis phenotype (Supplementary Fig. 3a-f) as *udu-gfp* RNA did (Supplementary Fig. 3g-i). These results demonstrate a specific role for Gon4l in axial extension during gastrulation.

**Gon4l regulates formation of the notochord boundary.** Time-lapse Nomarski (Fig. 2a, b) and confocal (Fig. 2c, d) microscopy of dorsal mesoderm in MZ*udu−/−* gastrulae revealed reduced definition and regularity of the boundary between axial and paraxial mesoderm, hereafter referred to as the notochord boundary, compared to WT (Fig. 2a, b, arrowheads). The ratio of the total/net length of notochord boundaries was significantly

higher in MZ*udu−/−* than in WT gastrulae at all-time points (Fig. 2e, two-way ANOVA $p < 0.0001$), indicative of decreased straightness. Interestingly, laminin was detected by immunostaining at the notochord boundary of both WT and MZ*udu−/−* embryos (Fig. 2f, g), indicating that MZ*udu* mutants form a bona fide boundary, albeit an irregular one. These results demonstrate that Gon4l is necessary for proper formation of the notochord boundary during gastrulation.

**Cell polarity and intercalation are reduced in MZ*udu* mutants.** In vertebrate gastrulae, C&E is achieved chiefly through ML intercalation of polarized cells that elongate and align their cell bodies with the ML embryonic axis[3,5,7,9]. To determine

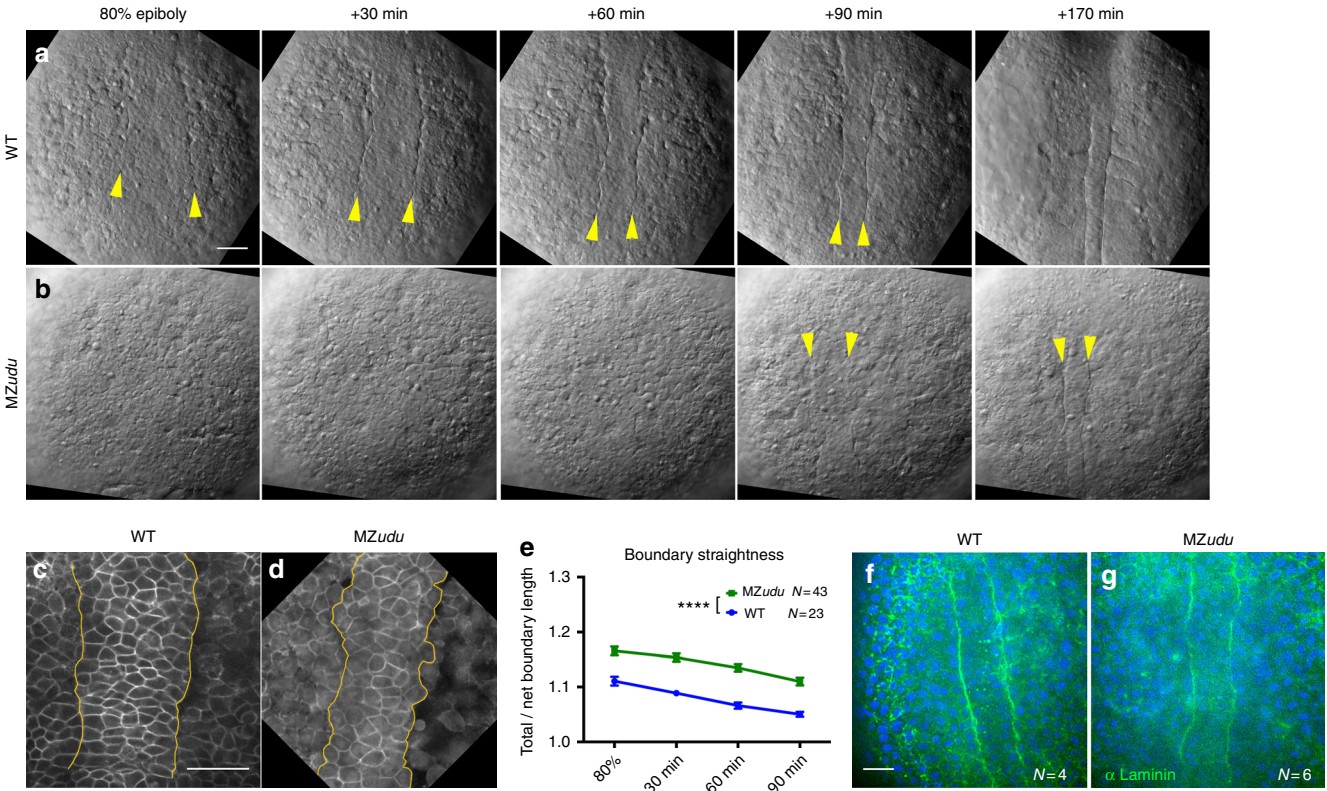

**Fig. 2** MZ*udu* mutant gastrulae exhibit irregular notochord boundaries. **a–b** Still images from live Nomarski time-lapse series of the dorsal mesoderm in WT (**a**) and MZ*udu*−/− embryos (**b**) at the time points indicated. Images are representative of over 40 MZ*udu*−/− gastrulae. Arrowheads indicate notochord boundaries. **c–d** Live confocal microscope images of representative WT (**c**, N = 23) and MZ*udu*−/− (**d**, N = 43) embryos expressing membrane Cherry. Yellow lines mark notochord boundaries. **e** Quantification of notochord boundary straightness in live WT and MZ*udu*−/− gastrulae throughout gastrulation. Symbols are means with SEM (two-way ANOVA, ****$p < 0.0001$). **f–g** Confocal microscope images of immunofluorescent staining for pan-Laminin in WT (**f**) and MZ*udu*−/− (**g**) embryos at two-somite stage. N indicates the number of embryos analyzed. Scale bars are 50 μm. Anterior is up in all images

whether cell polarity defects underlie reduced axis extension in MZ*udu*−/− gastrulae, we measured cell orientation (the angle of a cell's long axis with respect to the embryo's ML axis) and cell body elongation (length-to-width or aspect ratio (AR)) in confocal time-lapse series of fluorescent membrane-labeled WT (Fig. 3a–d) and MZ*udu*−/− (Fig. 3e–h) gastrulae. Given the irregular notochord boundaries observed in MZ*udu*−/− gastrulae (Fig. 2d, e), we examined the time course of cell polarization according to a cell's position with respect to the boundary, i.e., boundary-adjacent "edge" cells versus those one or two cell diameters away (hereafter −1 and −2, respectively), and so on (see Fig. 3). Most WT axial mesoderm cells at midgastrulation (80% epiboly) were largely ML oriented and somewhat elongated, with boundary-adjacent "edge" cells being the least well oriented (median angle = 24.6°) (Fig. 3a–d). However, at the end of the gastrula period 90 min later, edge cells became highly aligned and elongated (median angle = 13.6°) similar to internal cell rows (Fig. 3a–d). All MZ*udu*−/− axial mesoderm cells exhibited significantly reduced ML alignment (Fig. 3g, Kolmogorov–Smirnov test $p < 0.0001$), and elongation (Fig. 3h, Mann–Whitney test $p < 0.0001$) at 80% epiboly, but 90 min later only the edge cells remained less aligned than WT (median angle = 17.0°) (Fig. 3g), although AR of the edge and −1 cells remained reduced (Fig. 3h). These results indicate significantly reduced ML orientation of axial mesoderm cells in MZ*udu*−/− gastrulae, a defect that persisted only in boundary-adjacent cells at late gastrulation. Importantly, this reduction in ML cell polarity was accompanied by significantly fewer cell intercalation events within the axial

mesoderm of MZ*udu* mutants compared to WT (Fig. 3i–k, *T*-test $p < 0.05$). As ML intercalation is the key cellular behavior required for vertebrate C&E[3,5], we conclude that this is likely the primary cause of axial mesoderm extension defects in MZ*udu* mutants. We also observed significantly fewer mitoses in MZ*udu*−/− gastrulae (Fig. 3l–n, *T*-tests), consistent with reports of Z*udu* mutants[28]. Because decreased cell proliferation and the resulting reduced number of axial mesoderm cells were demonstrated to impair extension in zebrafish[31], this could also be a contributing factor. Together, reduction of ML cell polarity, ML cell intercalations, and cell proliferation provide a suite of mechanisms resulting in impaired axial extension in MZ*udu*−/− gastrulae. Combined with irregular notochord boundaries observed in MZ*udu* mutants, we hypothesize that this boundary provides a ML orientation cue that contributes to ML polarization of axial mesoderm cells, and that this cue is absent or reduced in MZ*udu*−/− gastrulae.

**Gon4l functions independently of PCP signaling**. Molecular control of ML cell polarity underlying C&E movements in vertebrate embryos is chiefly attributed to PCP and Gα12/13 signaling[6–10]. While zygotic loss of *udu* enhanced axial extension defects in *kny*^m818/m818^ PCP mutants (Fig. 1), it was unclear whether *udu* functions within or parallel to the PCP network. To address this, we generated compound MZ*udu*;Z*kny*^fr6/fr6^ (a nonsense/null *kny* allele[8]) mutants utilizing germline replacement as described above. Strikingly, these compound mutant

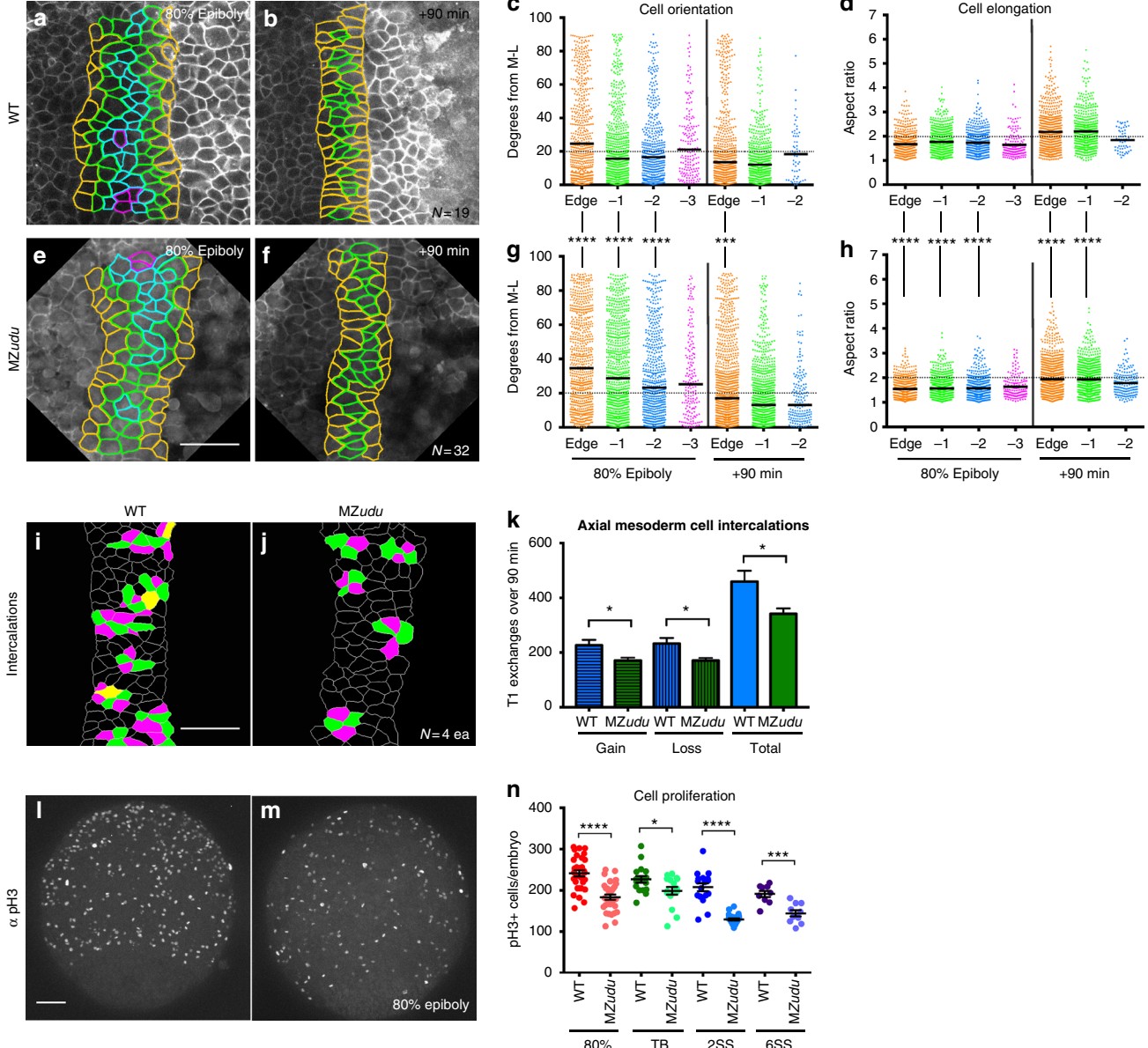

**Fig. 3** Mediolateral cell polarity and cell intercalations are reduced in the axial mesoderm of MZ*udu*−/− gastrulae. **a–b**, **e–f** Still images from live time-lapse confocal movies of the axial mesoderm in WT (**a**, **b**) and MZ*udu*−/− (**e**, **f**) gastrulae at the time points indicated. Cell outlines are colored according to a cell's position with respect to the notochord boundary. **c**, **g** Quantification of axial mesoderm cell orientation at 80% epiboly (left side) and +90 min (right side) time points. Each dot represents the orientation of the major axis of a single cell with respect to the embryonic ML axis and is colored according to that cell's position with respect to the notochord boundary (as in images to the left). Bars indicate median values. Asterisks indicate significant differences between WT and MZ*udu*−/− (Kolmogorov–Smirnov test, ***$p < 0.001$, ****$p < 0.0001$). **d**, **h** Quantification of axial mesoderm cell elongation at 80% epiboly (left side) and +90 min (right side) time points. Each dot represents the aspect ratio of a single cell and is color-coded as in **c**. Bars indicate mean values. Asterisks indicate significant differences between WT and MZ*udu*−/− (Mann–Whitney test, ****$p < 0.0001$). **i–j** Cell intercalations detected in the axial mesoderm of WT (**i**) and MZ*udu*−/− gastrulae (**j**). Cells gaining contacts with neighbors are green, cells losing contacts are magenta, and cells that both gain and lose contacts are yellow. **k** Quantification of cell intercalation events (T1 exchanges) in WT (blue bars) and MZ*udu*−/− gastrulae (green bars) over 90 min. N indicates the number of embryos analyzed. Bars are means with SEM (T-test, *$p < 0.05$). **l–m** 200 μm confocal Z projections of immunofluorescent staining for phosphorylated Histone H3 (pH3) in WT (**l**) and MZ*udu*−/− gastrulae (**m**) at 80% epiboly. Images are representative of eight independent trials. **n** Quantification of pH3+ cells/embryo at the stages indicated. Each dot represents a single embryo, dark lines are means with SEM (T-test, *$p < 0.05$, ***$p < 0.001$, ****$p < 0.0001$). Scale bars are 50 μm. Anterior/animal pole is up in all images

embryos were substantially shorter than single MZ*udu* or *kny*$^{fr6/fr6}$ mutants (also strict maternal *udu* mutants, which have no obvious phenotype) (Fig. 4a–d). Likewise, interference with *vangl2/tri* function in MZ*udu* mutants by injection of MO1-*vangl2* antisense morpholino oligonucleotide (MO)[32] also exacerbated axis extension defects of MZ*udu* mutants (Supplementary Fig. 4a–d). That reduced levels of PCP components *kny*

or *tri* enhanced short axis phenotypes resulting from complete *udu* deficiency provides evidence that Gon4l affects axial extension via a parallel pathway. Furthermore, expression domains of genes encoding Wnt/PCP signaling components *kny*, *tri*, and *wnt5* in MZ*udu*−/− gastrulae were comparable to WT (Supplementary Fig. 4e–l). Finally, we found that the asymmetric intracellular localization of Prickle (Pk)-GFP, a core PCP component,

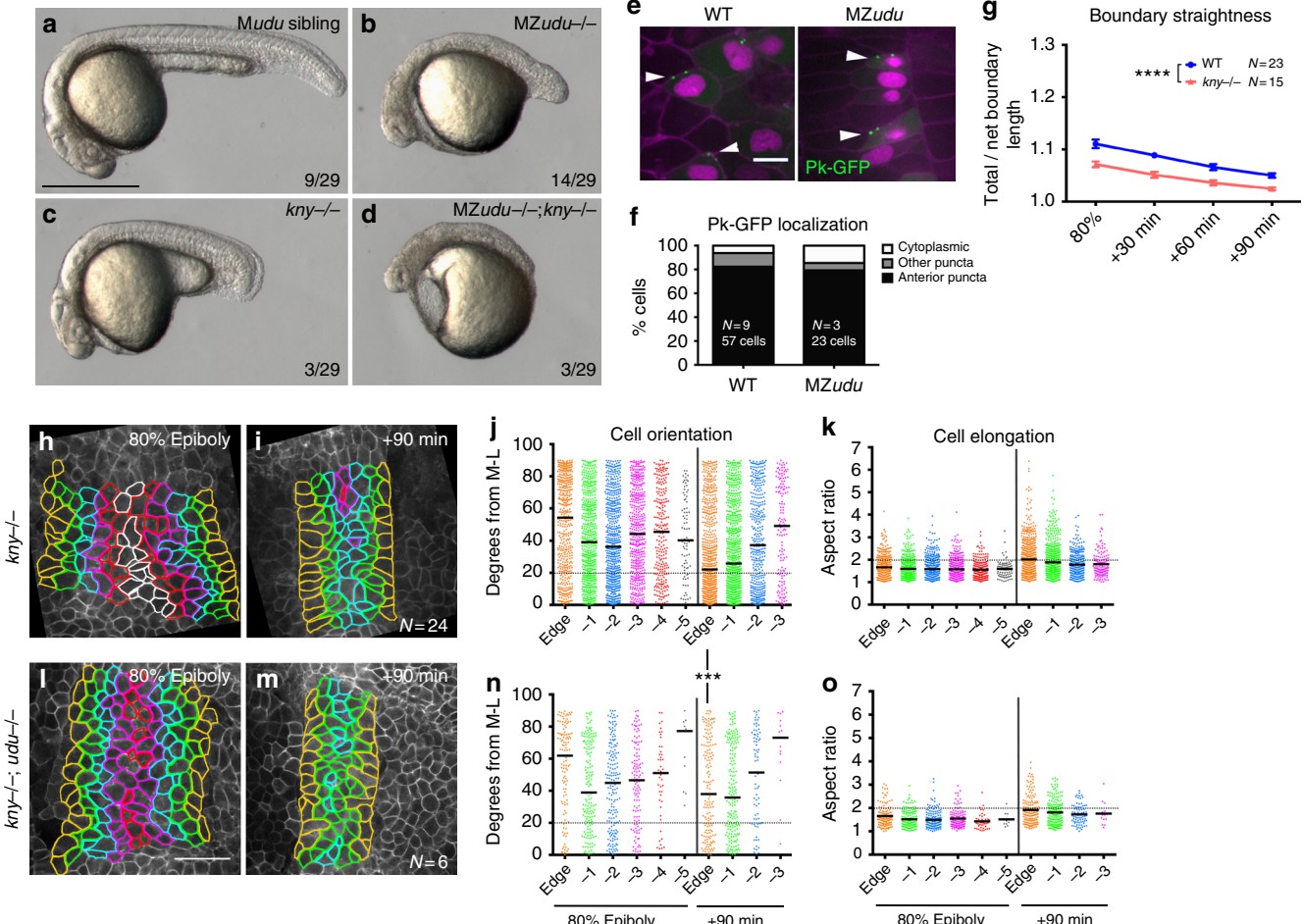

**Fig. 4** Gon4l regulates axis extension independent of PCP signaling. **a–d** Live embryos at 24 hpf resulting from a cross between a germline-replaced udu−/−;kny^fr6/+ female and an udu+/−;kny^fr6/+ male. Genotypes are indicated in the upper right corner, fractions indicate the number of embryos in the clutch with the pictured phenotype. **e** Mosaically expressed Prickle (Pk)-GFP in WT and MZudu−/− gastrulae. Arrowheads indicate anteriorly localized Pk-GFP puncta. Membrane Cherry marks cell membranes, nuclear-RFP marks cells injected with pk-gfp RNA. Images are representative of three independent experiments. **f** Quantification of Pk-GFP localization shown in **e** (chi-square, p = 0.07). **g** Quantification of notochord boundary straightness in WT and kny−/− gastrulae. Symbols are means with SEM (two-way ANOVA, ****p < 0.0001). **h, i, l, m** Still images from live time-lapse confocal movies of the axial mesoderm in kny−/− (**h–i**) and kny−/−;udu−/− (**l–m**) gastrulae at the time points indicated. Cell outlines are colored as in Fig. 3. **j, n** Quantification of axial mesoderm cell orientation as in Fig. 3, bars are median values (Kolmogorov–Smirnov test, ***p = 0.0003). **k, o** Quantification of axial mesoderm cell elongation as in Fig. 3, bars are mean values. N indicates number of embryos analyzed (and number of cells in **f**). Scale bar is 500 μm in **a–d**, 10 μm in **e**, 50 μm in **h–m**

to anterior cell membranes in WT gastrulae during C&E movements[33,34] was not affected in MZudu−/− gastrulae (Fig. 4e, f). This is consistent with intact PCP signaling in MZudu mutant gastrulae, and provides further evidence that Gon4l functions largely in parallel to PCP to regulate ML cell polarity and axis extension during gastrulation.

**Gon4l-dependent boundary cue cooperates with PCP signaling.** In addition to PCP signaling, notochord boundaries are required for proper C&E of the axial mesoderm in ascidian embryos[35] and are involved in the polarization of intercalating cells during Xenopus gastrulation[1]. Boundary defects observed in MZudu−/− gastrulae are not a common feature of mutants with reduced C&E, however, as notochord boundaries in kny^fr6/fr6 gastrulae were straighter than in WT (Fig. 4g). Consistent with cell polarity defects previously reported in kny mutants[8], all axial mesoderm cells failed to align ML within kny−/− embryos at 80% epiboly regardless of their position relative to the notochord boundary (Fig. 4h–k). After 90 min, however, kny−/− cells in the edge (and

to a lesser extent, −1) position attained distinct ML orientation (median angle = 22.1°)(Fig. 4j). Indeed, the nearer a kny−/− cell was to the notochord boundary, the more ML aligned it was likely to be. This suggests that the notochord boundary provides a ML orientation cue that is independent of PCP signaling and functions across approximately two cell diameters. Furthermore, this boundary-associated cue appears to operate later in gastrulation, whereas PCP-dependent cell polarization is evident by 80% epiboly. Importantly, distinct cell polarity phenotypes observed in MZudu−/− and kny−/− gastrulae provide further evidence that Gon4l functions in parallel to PCP signaling.

To assess how PCP signaling interacts with the proposed Gon4l-dependent boundary cue, we examined the polarity of axial mesoderm cells in compound zygotic kny−/−;udu−/− mutant gastrulae (Fig. 4l–o). As observed in single kny−/− mutant gastrulae[8] (Fig. 4h–k), both ML orientation and elongation of all axial mesoderm cells were reduced in double mutants at 80% epiboly, regardless of position with respect to the notochord boundary (Fig. 4n, o). By late gastrulation, however, edge cells in double mutant gastrulae failed to attain the ML alignment observed

in *kny−/−* mutants, but instead remained largely randomly oriented (median angle = 38.0°) (Fig. 4n). This exacerbation of cell orientation defects correlated with stronger axis extension defects in compound *kny−/−;udu−/−* compared to single *kny−/−* mutants (Fig. 1e). This supports our hypothesis that a Gon4l-dependent boundary cue regulates ML alignment of axial mesoderm cells independent of PCP signaling, and that these two mechanisms partially overlap in time and space to cooperatively polarize all axial mesoderm cells and promote axial extension.

**Loss of Gon4l results in large-scale gene expression changes**. As a nuclear-localized chromatin factor[18,36] (Supplementary Fig. 3j), Gon4l is unlikely to influence morphogenesis directly. To identify genes regulated by Gon4l with potential roles in morphogenesis, we performed RNA sequencing (RNA-seq) in MZ*udu−/−* and WT tailbud-stage embryos. Analysis of relative transcript levels revealed that more than 11% of the genome was differentially expressed ($p_{adj} < 0.05$, ≥twofold change) in MZ*udu* mutants (Fig. 5a). Of these ~2950 differentially expressed genes, 1692 exhibited increased expression in MZ*udu* mutants compared to 1259 with decreased expression (Supplementary data 1). Functional annotation analysis revealed that genes downregulated in MZ*udu−/−* gastrulae were enriched for ontology terms related to chromatin structure, transcription, and translation (Fig. 5b, c), while upregulated genes were enriched for terms related to biosynthesis, metabolism, and protein modifications (Fig. 5b, d). Notably, many of these misregulated genes are considered to have "housekeeping" functions, in that they are essential for cell survival. We further examined genes with plausible roles in morphogenesis, including those encoding signaling and adhesion molecules, and found the majority of genes in both classes were expressed at higher levels in MZ*udu−/−* gastrulae (Fig. 5e, f). This putative increase in adhesion was of particular interest given the tissue boundary defects observed in MZ*udu* mutants.

**DamID-seq identifies putative direct targets of Gon4l**. To determine genomic loci with which Gon4l protein associates, and thereby distinguish direct from indirect targets of Gon4l regulation, we employed DNA adenine methyltransferase (Dam) identification paired with next generation sequencing (DamID-seq)[20,21]. To this end, we generated Gon4l fused to *Escherichia coli* Dam, which methylates adenine residues within genomic regions in its close proximity[20]. Small equimolar amounts of RNA encoding a Myc-tagged Gon4l-Dam fusion or a Myc-tagged GFP-Dam control were injected into one-celled embryos, and genomic DNA was collected at tailbud stage (see Methods section). In support of this Gon4l-Dam fusion being functional, it localized to the nuclei of zebrafish embryos and partially rescued MZ*udu−/−* embryonic phenotypes (Supplementary Fig. 5a-k). Methylated, and therefore Gon4l-proximal, genomic regions were then selected using methylation specific restriction enzymes and adaptors, amplified to produce libraries, and sequenced. Because Dam is highly active, even an untethered version methylates DNA within open chromatin regions[20], and so libraries generated from embryos expressing GFP-Dam served as controls.

Unbiased genome-wide analysis of DamID reads revealed a significant enrichment of Gon4l-Dam over GFP-Dam in promoter regions, and a significant underrepresentation of Gon4l within intergenic regions (Fig. 6a, T-test $p < 0.05$, Fig. 6b). Although no global difference was detected within gene bodies (Fig. 6a), examination of individual loci revealed ~4500 genes and over 2300 promoters in which at least one region was highly Gon4l-enriched ($P_{adj} \leq 0.01$, ≥fourfold enrichment over GFP) (Fig. 6c–g, Supplementary Data 2, 3). Of these, ~1000 genes were co-enriched for Gon4l in both the promoter and gene body

(Fig. 6c, g). Levels of Gon4l association across a gene and its promoter were significantly correlated (Supplementary Fig. 5l, Spearmann correlation $p < 0.0001$), indicating co-enrichment or co-depletion for Gon4l at both gene features of many loci. Within gene bodies, we found robust enrichment specifically within 5′ untranslated regions (UTRs) (Fig. 6b), consistent with association of Gon4l at or near transcription start sites (Fig. 6d). Of the ~2950 genes differentially expressed in MZ*udu−/−* gastrulae, approximately 28% (812) were also enriched for Gon4l at the gene body (492), promoter (170), or both (150) (Fig. 6g), and will hereafter be described as putative direct Gon4l targets. *histh1*, for example, was among the most downregulated genes by RNA-seq in MZ*udu* mutants (Supplementary Data 1) and was highly enriched for Gon4l at both its promoter and gene body (Fig. 6c). By contrast, *tbx6* expression was also reduced in MZ*udu* mutants (Supplementary Data 1), but exhibited no enrichment of Gon4l over GFP controls (Fig. 6f). Approximately 35% and 53% of genes enriched for Gon4l in only the gene body or only the promoter, respectively, were positively regulated (i.e., downregulated in MZ*udu* mutants), as were 50% of genes co-enriched at both features (Fig. 6h). Furthermore, among these positively regulated genes, differential expression levels (the degree to which WT expression exceeded MZ*udu−/−*) correlated positively and significantly with Gon4l-enrichment levels across both gene bodies and promoters (Fig. 6h, Spearmann correlation $p < 0.01$). A similar correlation was not observed for negatively regulated genes, hence, the highest levels of Gon4l enrichment were associated with positive regulation of gene expression. These results implicate Gon4l as both a positive and negative regulator of gene expression during zebrafish gastrulation.

**Gon4l limits *itga3b* to promote boundary straightness**. We next examined our list of putative direct Gon4l target genes for those with potential roles in tissue boundary formation and/or cell polarity. *epcam*, which encodes Epithelial cell adhesion molecule (EpCAM), stood out because it was not only enriched for Gon4l by DamID (Fig. 7a) and upregulated in MZ*udu−/−* gastrulae (Fig. 5f, Supplementary Data 1), but was also identified in a *Xenopus* overexpression screen for molecules that disrupt tissue boundaries[37]. Furthermore, EpCAM negatively regulates non-muscle myosin activity[38], making it a compelling candidate. We also chose to examine *itga3b*, which encodes Integrinα3b, because as a component of a laminin receptor[39], it is an obvious candidate for a molecule involved in formation of a tissue boundary at which laminin is highly enriched (Fig. 2f). DamID revealed a region within the *itga3b* promoter at which Gon4l was highly enriched (Fig. 7b, c), and *itga3b* expression was increased in MZ*udu−/−* gastrulae by RNA-seq (Supplementary Data 1), which was also validated (along with *epcam*) by qRT-PCR (Fig. 7d, e). We found that overexpression of either *epcam* or *itga3b* by RNA injection into WT embryos recapitulated the irregular notochord boundaries (Fig. 7f, g) and reduced ML orientation and elongation of axial mesoderm cells (Supplementary Figs. 6d-g, 7c-f) seen in MZ*udu−/−* gastrulae. Conversely, while injection of an *epcam* MO (MO2-*epcam*[40], Supplementary Fig. 7b) had no effect on MZ*udu−/−* boundaries (Fig. 7h), injection of a translation-blocking *itga3b* MO (MO1-*itga3b*[41], Supplementary Fig. 6b) significantly improved boundary straightness in MZ*udu* mutants compared with control-injected siblings (Fig. 7i, two-way ANOVA $p < 0.0001$). This indicates that excess *itga3b* is largely responsible for reduced straightness of notochord boundaries in MZ*udu−/−* gastrulae. Furthermore, disruption of WT notochord boundaries by excess *itga3b* was abrogated by co-injection with a MO against *lama5* (MO3-*lama5*[42], Supplementary Fig. 6c) (Fig. 7j), which

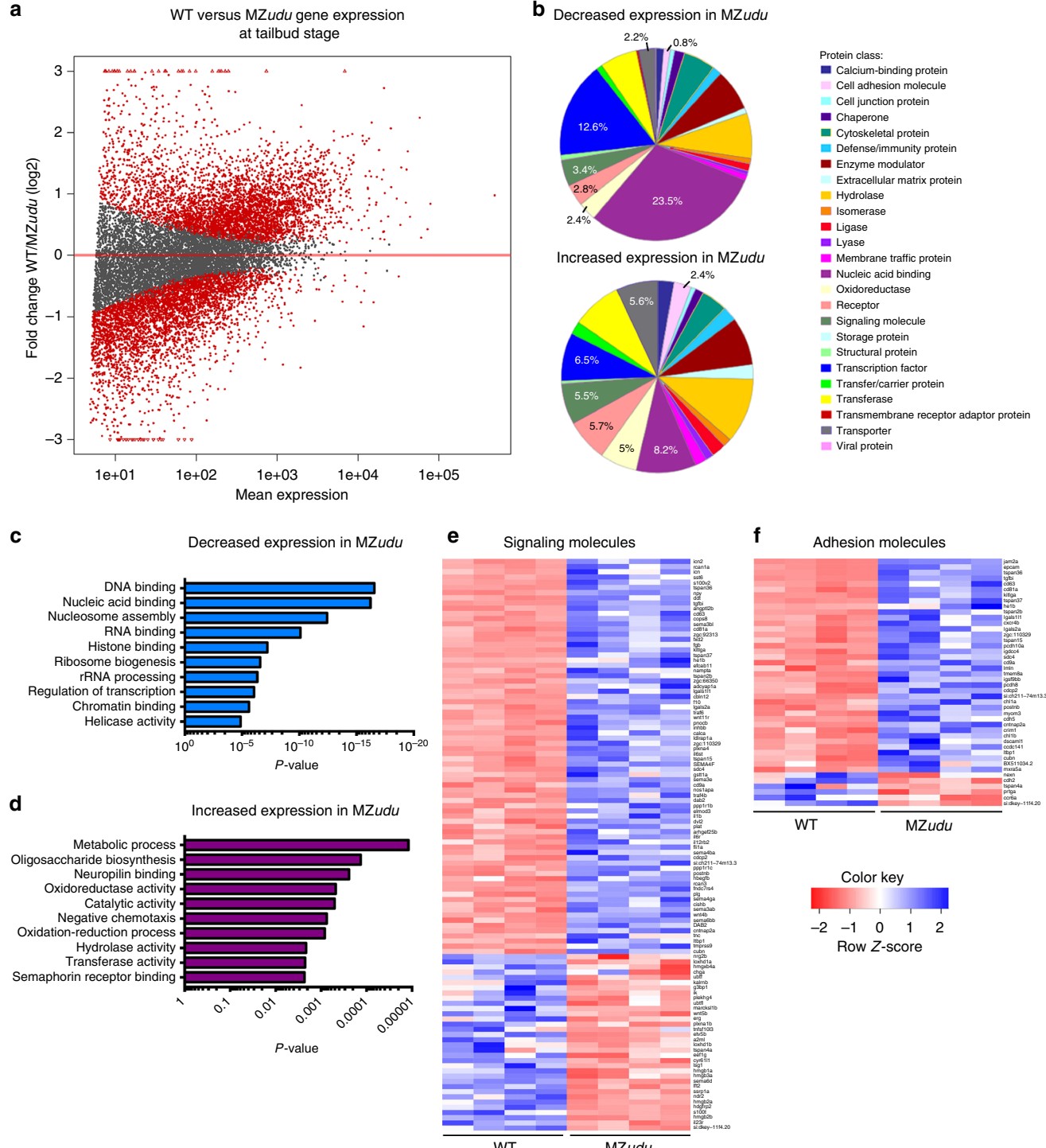

**Fig. 5** Loss of Gon4l results in large-scale gene expression changes. **a** Plot of gene expression changes in MZ*udu* mutants compared to WT at tailbud stage as assessed by RNA sequencing. Red dots indicate genes expressed at significantly different levels than WT ($p \leq 0.05$, at least twofold change in transcript level). **b** Protein classes encoded by differentially expressed genes with decreased (top graph) or increased (bottom graph) expression in MZ*udu* mutants. Percentages indicate the number of genes within a given class/total number of genes with increased or decreased expression, respectively. **c–d** P-values of the top ten most enriched gene ontology (GO) terms among genes with decreased (**c**) or increased (**d**) expression in MZ*udu*−/− gastrulae compared to WT. **e–f** Heat maps of differentially expressed genes annotated as encoding signaling (**e**) or adhesion molecules (**f**). The four columns represent two biological and two technical replicates for each WT and MZ*udu*−/−

encodes its ligand Lamininα5[41], indicating that this effect is ligand dependent. Both *itga3b* and *lama5* are expressed within the axial mesoderm of WT zebrafish gastrulae[43,44]. Moreover, although an Itgα3b-GFP fusion was not enriched at WT notochord boundaries, it became increasingly localized to the plasma membrane of axial mesoderm cells (but not presomitic mesoderm or neuroectodermal cells) as gastrulation proceeded (Fig. 7k–m). This implies that intracellular localization (and likely function) of Integrinα3b is regulated in a stage- and tissue-dependent manner, as was reported for other Integrins during *Xenopus* gastrulation[45].

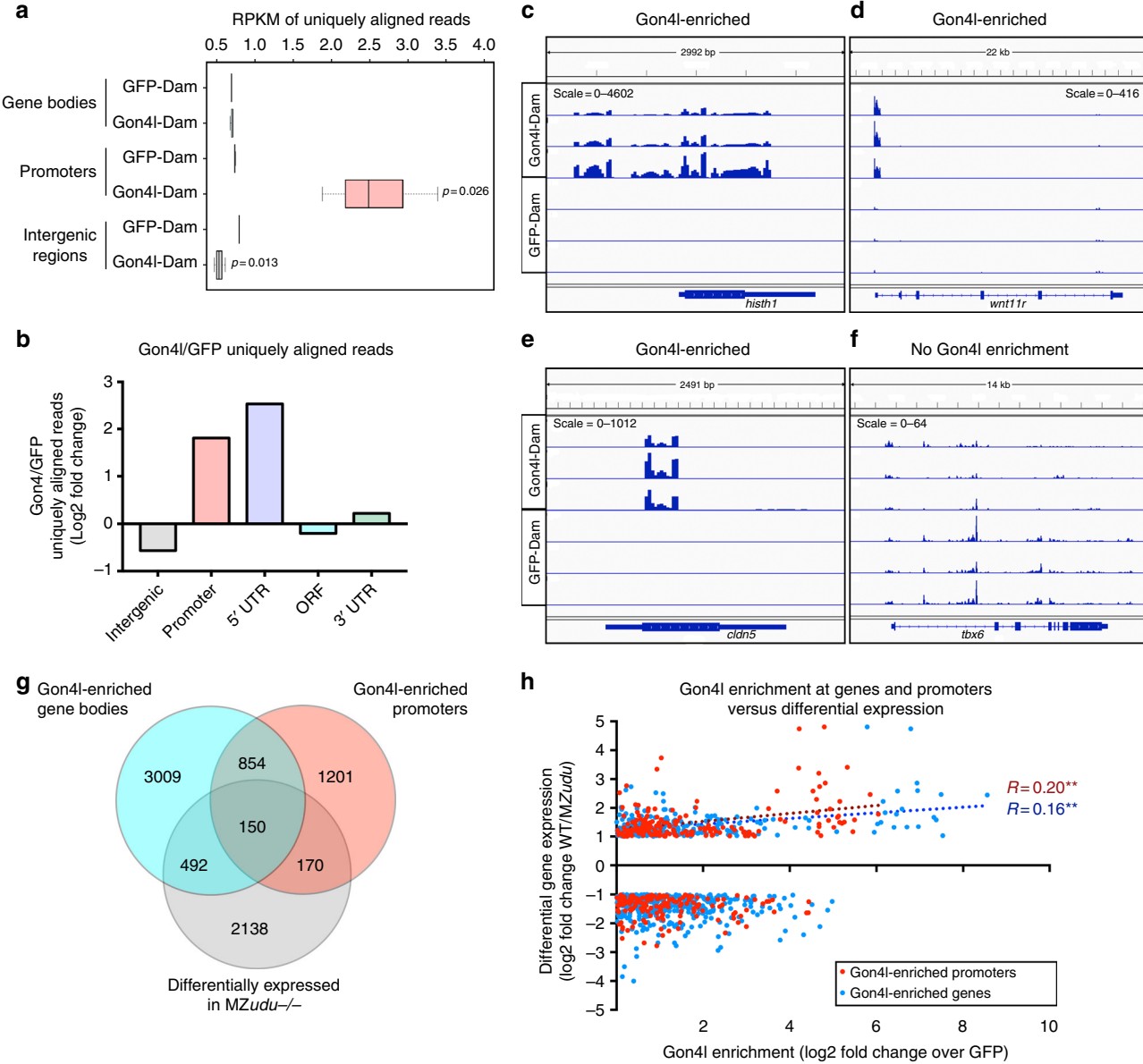

**Fig. 6** DamID-seq identifies putative direct targets of Gon4l during gastrulation. **a** Box plot of normalized uniquely aligned DamID reads at each of three categories of genomic regions. Center lines are medians, box limits are upper and lower quartiles, whiskers are highest and lowest values. *P*-values indicate significant differences between Gon4l-Dam samples and GFP-Dam controls (*T*-test). **b** Fold change (log2) Gon4l-Dam over GFP-Dam reads (RPKM) at each of five gene features. **c**–**f** Genome browser tracks of Gon4l-Dam and GFP-Dam association at the *histh1* (**c**), *wnt11r* (**d**), *cldn5* (**e**), and *tbx6* (**f**) loci. Scale is number of reads. Each track represents one biological replicate at tailbud stage. **g** Venn diagram of genes with regions of significant Gon4l enrichment within gene bodies (blue) or promoters (pink), and genes differentially expressed in MZ*udu*−/− gastrulae (gray). **h** Correlation of Gon4l enrichment levels across gene bodies (blue dots) and promoters (red dots) with relative transcript levels of genes differentially expressed in MZ*udu* mutants. A positive correlation was detected between increased expression in WT relative to MZ*udu* mutants and Gon4l enrichment in both gene bodies (Spearman correlation *p* = 0.0095) and promoters (*p* = 0.0098). Dotted lines are linear regressions of these correlations

Finally, to determine whether excess *itga3b* was sufficient to phenocopy the *kny*−/−;*udu*−/− double mutant phenotype identified in our synthetic screen (Fig. 1), we overexpressed *itga3b* in *kny*−/− embryos and found that it exacerbated the short axis phenotype of these PCP mutants (Fig. 7n–p), an effect not produced by injection of control *GFP* RNA. Together, these results indicate that negative regulation of *itga3b* expression by Gon4l is essential for proper notochord boundary formation and axis extension in zebrafish gastrulae. Interestingly, ML cell polarity defects were not suppressed in MZ*udu*−/− embryos injected with *itga3b* (or *epcam*) MO compared to control-injected mutant siblings (Supplementary Figs. 6h-o, 7g-n), despite improved

boundary straightness. This implies that other Gon4l-dependent boundary properties are involved, and/or that loss of *udu* affects ML polarity cell-autonomously, for example, by making cells unable to respond to the boundary-associated polarity cue.

**Loss of Gon4l and excess *epcam* reduce boundary tension.** In WT zebrafish and *Xenopus* embryos, notochord boundaries straighten over time (Fig. 2e) and accumulate myosin[46], implying that they are under tension. Because boundaries of MZ*udu* mutants are less straight than those of WT gastrulae, we hypothesized that tension at these boundaries is reduced. To test this,

we laser-ablated interfaces between axial mesoderm cells in live WT and MZ*udu*−/− gastrulae and recorded recoil of adjacent cell vertices as a measure of tissue tension[47] (Fig. 8a, b). Cell interfaces were classified according to an established convention

as V junctions (actively shrinking to promote cell intercalation), T junctions (not shrinking)[47,48] or Edge junctions (falling on and thus comprising the notochord boundary). We found that recoil distances at WT Edge junctions were significantly greater than

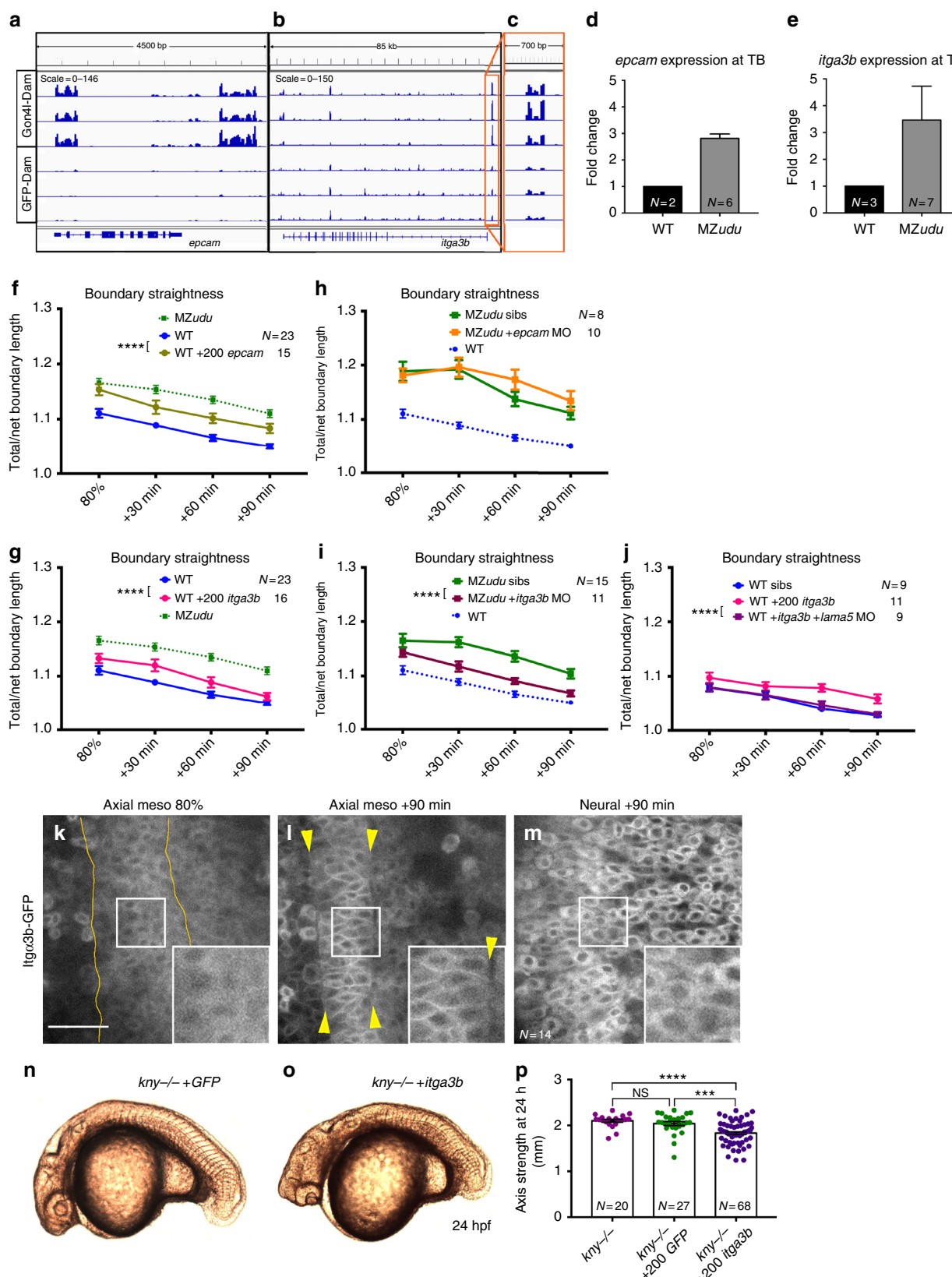

that of WT V (Kruskal–Wallis test $p = 0.0001$) or T junctions (Kruskal–Wallis test $p = 0.0026$), demonstrating that the notochord boundary is under greater tension than the rest of the tissue (Fig. 8c–e). Consistent with our hypothesis, recoil distance was significantly smaller in MZ*udu*−/− than in WT gastrulae for all classes of junctions (Fig. 8c–e, two-way ANOVA $p < 0.01$), and this decrease was largest and most significant in Edge junctions (Fig. 8c, two-way ANOVA $p < 0.0001$).

Because excess *epcam* and *itga3b* were sufficient to reduce boundary straightness (Fig. 7), we next tested whether *epcam* and *itga3b* overexpression could likewise affect boundary tension. Injection of WT embryos with *epcam* RNA was sufficient to lower tissue tension at the notochord boundary and throughout the axial mesoderm (Fig. 8c–e). Curiously, although we found *itga3b* to be largely causative of reduced boundary straightness in MZ*udu* mutants (Fig. 7), its overexpression did not produce a similar reduction in tension (Fig. 8c–e), implying a tension-independent role for Itgα3b in boundary straightness. Accordingly, treatment of *itga3b*-overexpressing WT embryos with Calyculin A, a myosin phosphatase inhibitor that increases myosin contractility[49] and notochord boundary tension (Supplementary Fig. 8a), did not restore their boundary straightness despite increased tension (Fig. 8f, Supplementary Fig. 8c). By contrast, Calyculin A treatment significantly increased boundary tension and straightness in *epcam*-overexpressing WT embryos (Fig. 8g, Supplementary Fig. 8d, two-way ANOVA), indicating a tension-dependent role for EpCAM in boundary formation. Consistent with its previously described inhibition of myosin activity[38], these data support a role for EpCAM as a negative regulator of myosin-dependent tissue tension at the notochord boundary. Notably, treatment of MZ*udu* mutants with Calyculin A did not restore boundary straightness or axial mesoderm cell polarity (Supplementary Fig. 8b, e-h), demonstrating that increasing tension alone was not sufficient to improve these phenotypes. Given that the effect of EpCAM on the boundary is largely tension dependent, this likely explains why the *epcam* MO failed to improve boundary straightness in MZ*udu* mutants (Fig. 7h). Together these results implicate excess EpCAM and Integrinα3b as key molecular defects underlying reduced boundary tension and reduced boundary straightness, respectively, observed in MZ*udu*−/− gastrulae. We propose a model whereby Gon4l limits expression of these adhesion molecules to ensure proper formation of the notochord boundary, which together with additional boundary-independent roles of Gon4l cooperates with PCP signaling to promote ML cell polarity underlying C&E gastrulation movements (Fig. 8h).

## Discussion

Substantial advances have been made in defining signaling pathways that regulate gastrulation cell behaviors and shape the vertebrate body plan[6,7,9,10], but epigenetic control of these morphogenetic processes remains largely unexplored. Here we have

described the conserved chromatin factor Gon4l as a regulator of polarized cell behaviors underlying axis extension during zebrafish gastrulation. We identified a large number of Gon4l target genes, including many with known or predicted roles in morphogenesis, and linked misregulation of a subset of these genes to specific morphogenetic defects. Because Gon4l does not bind DNA directly, we predict that Gon4l-enriched genomic loci are direct targets of chromatin modifying protein complexes with which Gon4l associates[36]. Only a fraction of the thousands of Gon4l-enriched genes and promoters exhibited corresponding changes in gene expression during gastrulation, which may reflect one or more possible scenarios. Either Gon4l does not alter expression of all loci with which it associates, loci recently occupied by Gon4l do not yet reflect changes in transcript levels, or because DamID provides a "history" of Gon4l association, both formerly and currently occupied loci are represented in our data. We also identified loci at which Gon4l was depleted compared to GFP-Dam controls (Supplementary Fig. 5), and speculate they represent open chromatin regions with which Gon4l does not associate. Surprisingly, although a larger number of putative Gon4l direct target genes were negatively regulated (i.e., upregulated in MZ*udu* mutants), the highest levels of Gon4l enrichment were correlated with positive regulation of gene expression (Fig. 6). This argues against Gon4l acting strictly as a negative regulator of gene expression, a role assigned to it based on in vitro evidence and thought to be mediated by its interactions with Histone deacetylases[36]. Our data instead indicate that Gon4l acts as both a positive and negative regulator of gene expression during zebrafish gastrulation, implying context-specific interactions with multiple epigenetic regulatory complexes.

Phenotypes caused by complete *udu* deficiency are conspicuously pleiotropic, but our studies point to remarkably specific roles for *udu* in gastrulation morphogenesis. Loss of *udu* function reduced tissue extension without affecting mesendoderm internalization, epiboly, prechordal plate migration, or convergence (see below). Moreover, the dorsal gastrula organizer and all three germ layers were specified (Fig. 1, Supplementary Fig. 2), indicating that MZ*udu* mutants do not suffer from a general delay or arrest of development; rather Gon4l regulates a specific subset of gastrulation cell behaviors, including ML cell polarity and intercalation in the axial mesoderm. Importantly, the role of Gon4l in these processes is independent of PCP signaling as supported by several lines of evidence (Fig. 4, Supplementary Fig. 4). Additionally, whereas PCP mutants exhibited reduced convergence[7,8], evidenced by a larger number of cell rows in *kny*−/− axial mesoderm (Fig. 4), MZ*udu* mutants contained a normal number of axial mesoderm cell rows (Fig. 3), indicating no obvious convergence defect. Despite these apparently parallel functions, RNA-seq and DamID-seq experiments revealed that Gon4l regulates expression of some PCP genes, including *wnt11*, *wnt11r*, *prickle1a*, *prickle1b*, *celsr1b*, *celsr2*, and *fzd2* (Supplementary Data 1-3). However, functional redundancies within the PCP network[50,51] likely allow for intact PCP signaling in MZ*udu*

**Fig. 7** Gon4l regulates notochord boundary straightness by limiting *itga3b* expression. **a–c** Genome browser tracks of Gon4l-Dam and GFP-Dam association at the *epcam* (**a**) and *itga3b* (**b**) locus. An expanded view of the *itga3b* promoter is shown in **c**. **d–e** Quantitative RT-PCR for *epcam* (**d**) and *itga3b* (**e**) in WT and MZ*udu*−/− embryos at tailbud stage. *N* indicates the number of independent clutches tested with technical triplicates of each, bars are means with SEM. **f–g** Quantification of notochord boundary straightness in WT *epcam* (**f**) or *itga3b* (**g**) overexpressing embryos (two-way ANOVA, ****$p < 0.0001$). **h–i** Notochord boundary straightness in MZ*udu*−/− *epcam* (**h**) or *itga3b* (**i**) morphants and sibling controls (two-way ANOVA, ****$p < 0.0001$). **j** Quantification of notochord boundary straightness in WT *itga3b*-overexpressing embryos with or without *lama5* MO (two-way ANOVA, ****$p < 0.0001$). *N* indicates the number of embryos analyzed, symbols are means with SEM. **k–m** Itgα3b-GFP localization in WT embryos in the tissues and at the time points indicated. Insets are enlarged from regions in white squares. Yellow lines and arrowheads mark notochord boundaries. Images are representative of three independent trials. Scale bar is 50 μm. **n–o** Live images of *kny*−/− embryos at 24 hpf injected with 200 pg *GFP* (**n**) or 200 pg *itga3b* (**o**) RNA. Images are representative of four independent experiments. **p** Quantification of axis length of injected *kny*−/− embryos at 24 hpf. Each dot represents one embryo, bars are means with SEM (*T*-tests, ***$p < 0.001$, ****$p < 0.0001$). *N* indicates number of embryos analyzed

mutants despite misregulation of some PCP components. Notably, gastrula morphology and cell polarity defects in MZ*udu* mutants were also distinct from ventralized and dorsalized patterning mutants with impaired C&E[11,12].

Modulation of adhesion at tissue boundaries has been implicated as a driving force of cell intercalation[46,52], and indeed genes annotated as encoding adhesion molecules tended to be expressed

at higher levels in MZ*udu*−/− gastrulae (Fig. 5). Two of these, *itga3b* and *epcam*, exhibited increased expression in MZ*udu*−/− gastrulae by RNA-seq and qRT-PCR, were identified as putative direct Gon4l targets by DamID (Fig. 7), and were each sufficient to reduce notochord boundary straightness and ML cell polarity when overexpressed in WT embryos (Fig. 7, Supplementary Figs. 6, 7). Furthermore, decreasing levels of Integrinα3b (but not

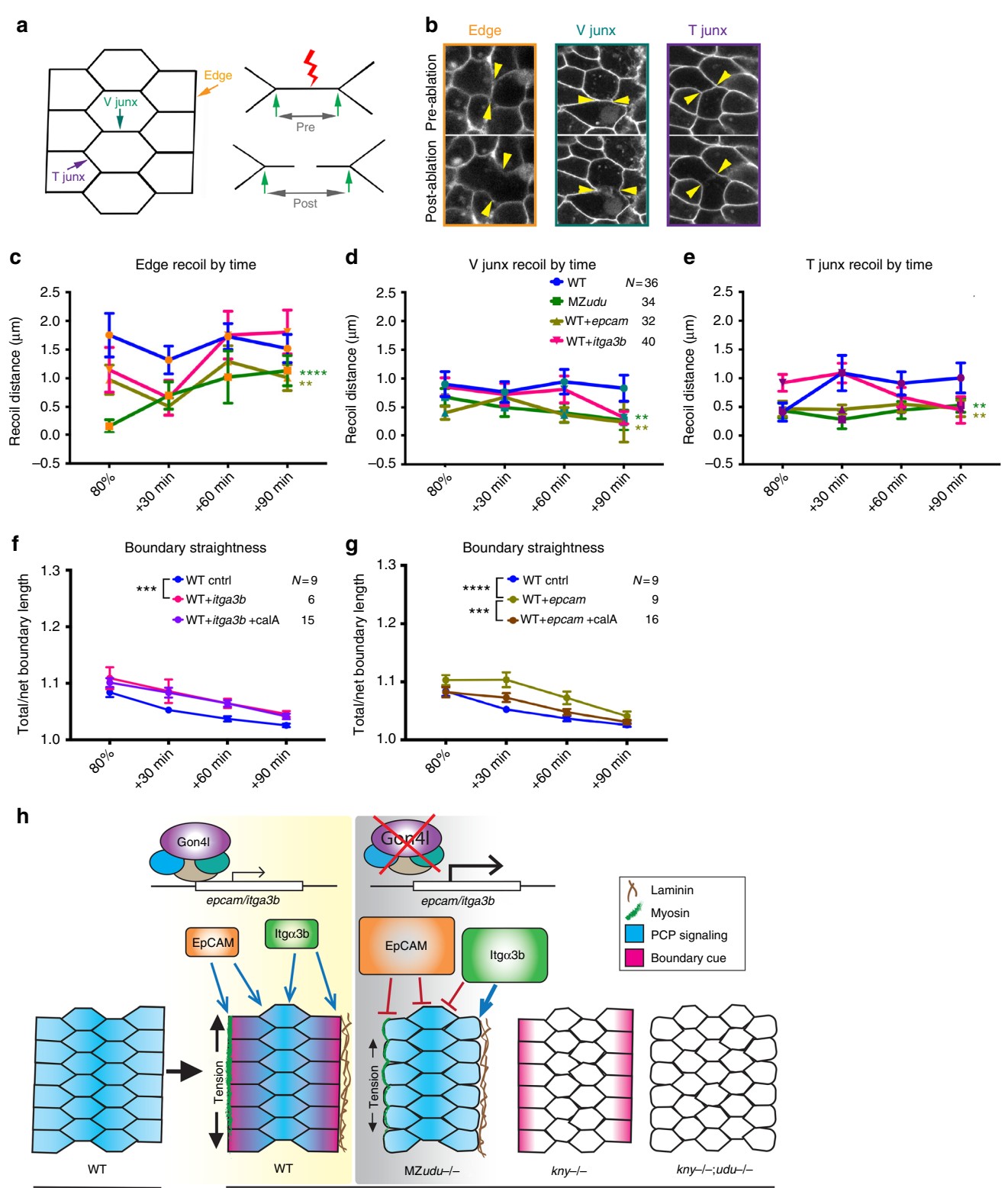

 

EpCAM) improved boundary straightness in MZ*udu* mutants (Fig. 7), while overexpression of *epcam* (but not *itga3b*) was sufficient to reduce tissue tension at the notochord boundary of WT embryos (Fig. 8). In addition, increasing tension with Calyculin A in *epcam*-overexpressing (but not *itga3b* overexpressing) WT embryos normalized their notochord boundary straightness (Fig. 8). These observations support a model in which mis-regulation of each of these molecules contributes to distinct cellular defects in MZ*udu* mutants: excess *itga3b* disrupts boundary straightness in a tension-independent manner via interactions with its laminin ligand, while EpCAM negatively regulates myosin-dependent boundary tension. EpCAM was similarly shown to negatively regulate non-muscle myosin contractility in *Xenopus* gastrulae[38], where experimental perturbation of myosin activity disrupts notochord boundary formation[46].

Links between myosin-dependent tissue tension, boundary formation, and cell intercalation are well established. In addition to *Xenopus* notochord boundaries[46], myosin accumulation increases tension at compartment boundaries in *Drosophila*[53], which was shown to bias cell intercalations[54]. We found that decreased boundary straightness in *epcam*-overexpressing WT gastrulae could be rescued by pharmacologically increasing tissue tension, demonstrating a causal relationship between tension and boundary straightness (Fig. 8). However, increasing tissue tension was not sufficient to improve boundary straightness in MZ*udu*−/− gastrulae (Supplementary Fig. 8), implying additional molecular bases for this phenotype, namely excess *itga3b*. Furthermore, restoration of boundary straightness (with *itga3b* MO) in MZ*udu*−/− gastrulae was not sufficient to normalize ML cell orientation, raising questions as to the functional relationship between these elements in this context. We speculate that increased boundary straightness and tension together may be required to promote ML polarity of boundary-adjacent cells. Alternatively, improved boundary straightness in MZ*udu*−/− gastrulae may have restored the polarity cue, but additional gene expression changes rendered mutant cells unable to respond to it. Although it is not yet clear whether and how boundary straightness and tension affect the polarity of adjacent cells, evidence from this study and others strongly implicate the notochord boundary in ML polarization of cells undergoing C&E[1,35]. Namely, within *kny*−/− PCP mutants, only boundary-adjacent edge cells became ML aligned during late gastrulation. Together with the additive cell polarity defects observed in compound *kny*−/−;*udu*−/− mutants, this is strong genetic evidence that the boundary provides a Gon4l-dependent, PCP-independent cell polarity cue (Fig. 4).

In this study of zebrafish Gon4l, we have begun to dissect the logic of epigenetic regulation of gastrulation cell behaviors by revealing a key role for this chromatin factor in limiting expression of specific genes with specific morphogenetic consequences. This has important developmental implications, as C&E gastrulation movements are sensitive to both gain and loss of gene function[7,10]. We propose that by negatively regulating Integrinα3b and EpCAM levels, Gon4l promotes development of the anteroposteriorly aligned notochord boundary and influences ML

polarity and intercalation of axial mesoderm cells. In cooperation with PCP signaling, this cue coordinates ML cell polarity with embryonic patterning to drive AP embryonic axis extension (Fig. 8h).

## Methods

**Zebrafish strains and embryo staging.** Adult zebrafish were raised and maintained according to established methods[55] in compliance with standards established by the Washington University Animal Care and Use Committee. Embryos were obtained from natural mating and staged according to morphology as described[56]. All studies on WT were carried out in AB* or AB*/Tübingen backgrounds. Additional lines used include *kny*[m818], *kny*[fr68], *udu*[sq118], and *udu*[vu66] (this work). Embryos of these strains generated from heterozygous intercrosses were genotyped by PCR after completion of each experiment. Germline-replaced fish were generated by the method described in ref. [27]. Briefly, donor embryos from *udu*[vu66/+] intercrosses or females with *udu*[vu66/vu66] germline were injected with synthetic RNA encoding *GFP-nos1-3'UTR*[57], and WT host embryos were injected with a MO against *dead end1* (MO1-*dnd1*)[27] to eliminate host germ cells. Cells were transplanted from the embryonic margin of donor blastulae to the embryonic margin of hosts at sphere stage, and both hosts and donors were cultured in agarose-coated plates. Host embryos were screened for GFP + germ cells at 36–48 hpf, and the genotype of corresponding donors was determined by phenotype. All putative *udu*[vu66/vu66] germline hosts were raised to adulthood and confirmed by crossing to *udu*[vu66/+] animals prior to use in experiments. Fish were chosen from their home tank to be crossed at random. The resulting embryos were also chosen from the dish at random for injection and inclusion in experiments. No fewer than six (but often many more) embryos of each condition or genotype were analyzed in at least three independent trials for each experiment in which measurements were made, except where noted. These rare cases reflect difficulty in obtaining sufficient MZ*udu*−/− embryos due to limited availability and productivity of germline-replaced females.

**Synthetic mutant screening.** WT male fish were mutagenized by the chemical mutagen ENU as described in ref. [22]. Briefly, adult male fish were incubated four to six times, for 1 h each, in a solution of 3.5 mM ENU at 21 °C. After treatment, fish were allowed to recover for 2–3 h in 10 mg/L 3-aminobenzoic acid ethyl ester (MESAB) at 17–19 °C and an additional day without MESAB before returning to the circulating water system. Approximately 3 weeks after mutagenesis, these males were outcrossed to WT females to produce F1 families. F2 families were obtained by crossing F1 fish with fish homozygous for the hypomorphic *knypek* allele *kny*[m818] (rescued by injection with synthetic *kny* WT RNA). F3 embryos obtained from F2 cross were screened by morphology at 12 and 24 hpf to identify recessive enhancers of the *kny*[m818/m818] short axis mutant phenotype[8].

**Positional cloning.** We employed the positional cloning approach using a panel of CA simple sequence length polymorphism markers representing 25 linkage groups[25] to map the *vu66* mutation to chromosome 16 between the markers of z17403 and z15431. Given phenotypic similarities between *vu66/vu66* and the *udu*[sq1/sq1] mutant phenotype[18], we sequenced *udu* cDNA from 24 hpf *vu66/vu66* mutant embryos revealing a T2261A transversion that is predicted to create Y753STOP nonsense mutation. We designed a dCAPS[58] marker for the *vu66* mutation and confirmed that no recombination occurred in 810 *vu66/vu66* homozygous embryos.

**Microinjection.** One-celled embryos were aligned within agarose troughs generated using custom-made plastic molds and injected with 1–3 pL volumes using pulled glass needles. Synthetic mRNAs for injection were made by in vitro transcription from linearized plasmid DNA templates using Invitrogen mMessage mMachine kits. Doses of RNA per embryo were as follows: 100 pg *membrane Cherry*, 50 pg *membrane GFP*, 25 pg *udu-gfp*, 200 pg *epcam*, 200 pg *itga3b*, 50 pg *itga3b-GFP*, 1 pg *gfp-dam-myc*, 3 pg *udu-dam-myc* for DamID experiments, 20 pg *gfp-dam-myc* for MZ*udu*−/− rescue, and 150 pg *GFP-nos1-3'UTR* for germline transplantation. To assess Pk-GFP localization, embryos were injected at one-cell stage with *membrane Cherry* RNA, then injected with 15 pg *Drosophila prickle-GFP* and 20 pg *H2B-RFP* RNAs into a single blastomere at 16-cell stage as

**Fig. 8** Loss of Gon4l and excess *epcam* reduce notochord boundary tension. **a** Diagram of laser ablation experiments to measure tension at axial mesoderm cell interfaces. **b** Still images from confocal time-lapse movies of each of the three types of cell interfaces (Edge, V junctions, and T junctions) before and after laser ablation. Arrowheads indicate cell vertices adjacent to the ablated interface. Images are representative of 36 independent experiments. **c–e** Quantification of cell vertex recoil distance immediately after laser ablation of Edge (**c**), V junction (**d**), and T junction (**e**) interfaces at the time points indicated in WT, MZ*udu*−/−, WT *epcam* overexpressing, and WT *itga3b*-overexpressing gastrulae. Symbols are means with SEM. Asterisks are colored according to key and indicate significant differences compared to WT controls (two-way ANOVA, **** $p < 0.0001$, ** $p < 0.01$). **f–g** Quantification of notochord boundary straightness in WT *itga3b* (**f**) or *epcam* (**g**) overexpressing embryos with or without Calyculin A (two-way ANOVA, **** $p < 0.0001$, *** $p < 0.001$). N indicates the number of embryos analyzed, symbols are means with SEM. **h** Graphical model of the roles of Gon4l and PCP signaling in regulating ML cell polarity of axial mesoderm cells and notochord boundary formation

described[33,34]. Injections of MOs were carried out as for synthetic RNA. Doses of MOs per embryo were as follows: 3 ng MO1-*dnd1*[27], 4 ng MO1-*tri/vangl2*[32], 1 ng MO2-*epcam*[40], 2 ng MO1-*itga3b*[41], and 1 ng MO3-*lama5*[42].

**WISH.** Antisense riboprobes were transcribed using NEB T7 or T3 RNA polymerase and labeled with digoxygenin (DIG) (Roche). WISH was performed according to ref. [59]. Briefly, embryos were fixed overnight in 4% paraformaldehyde (PFA) in phosphate buffered saline (PBS), rinsed in PBS+0.1% tween 20 (PBT), and dehydrated into methanol. Embryos were then rehydrated into PBT, incubated for at least 2 h in hybridization solution with 50% formamide (in 0.75 M sodium chloride, 75 mM sodium citrate, 0.1% tween 20, 50 μg/mL heparin (Sigma), and 200 μg/mL tRNA) at 70 °C, then hybridized overnight at 70 °C with antisense probes diluted approximately 1 ng/μl in hybridization solution. Embryos were washed gradually into 2X SSC buffer (0.3 M sodium chloride, 30 mM sodium citrate), and then gradually from SSC to PBT. Embryos were blocked at room temperature for several hours in PBT with 2% goat serum and 2 mg/mL bovine serum albumin (BSA), then incubated overnight at 4 °C with anti-DIG antibody (Roche #11093274910) at 1:5000 in block. Embryos were rinsed extensively in PBT, and then in staining buffer (PBT+100 mM Tris pH 9.5, 50 mM MgCl$_2$, and 100 mM NaCl) prior to staining with BM Purple solution (Roche).

**Immunofluorescent staining.** Embryos were fixed in 4% PFA, rinsed in PBT, digested briefly in 10 μg/mL proteinase K, refixed in 4% PFA, rinsed in PBT, and blocked in 2 mg/mL BSA + 2% goat serum in PBT. Embryos were then incubated overnight in rabbit anti-Laminin (Sigma L9393) at 1:200, mouse anti-Myc (Cell Signaling 2276) at 1:1000, or rabbit anti-phospho histone H3 (Upstate 06-570) at 1:500 in blocking solution, rinsed in PBT, and incubated overnight in Alexa Fluor 488 anti-Rabbit IgG, 568 anti-Rabbit, or 568 anti-Mouse (Invitrogen) at 1:1000 in PBT. Embryos were co-stained with 4′,6-Diamidino-2-Phenylindole, Dihydrochloride and rinsed in PBT prior to mounting in agarose for confocal imaging.

**TUNEL staining.** Terminal deoxynucleotidyl transferase (TdT) dUTP Nick-End Labeling (TUNEL) staining to detect apoptosis was carried out according to the instructions for the ApopTag Peroxidase in situ apoptosis detection kit (Millipore) with modifications. Briefly, embryos were fixed in 4% PFA, digested with 10 μg/mL proteinase K, refixed with 4% PFA, and post-fixed in chilled ethanol:acetic acid 2:1, rinsing in PBT after each step. Embryos were incubated overnight with TdT and rinsed in stop/wash buffer, then blocked, incubated with anti-DIG antibody, and stained in Roche BM Purple staining solution.

**Microscopy.** Live embryos expressing fluorescent proteins or fixed embryos subjected to immunofluorescent staining were mounted in 0.75% low-melt agarose in glass bottomed 35-mm petri dishes for imaging using a modified Olympus IX81 inverted spinning disc confocal microscope equipped with Voltran and Cobolt steady-state lasers and a Hamamatsu ImageM EM CCD digital camera. For live time-lapse series, 60 μm z-stacks with a 2 μm step were collected every three to ten minutes (depending on the experiment) for 3 h using a ×40 dry objective lens. Embryo temperature was maintained at 28.5 °C during imaging using a Live Cell Instrument stage heater. When necessary, embryos were extracted from agarose after imaging for genotyping. For immunostained embryos, 200 μm z-stacks with a 1 or 2 μm step were collected using a ×10 or ×20 dry objective lens, depending on the experiment. Bright field and transmitted light images of live embryos and in situ hybridizations were collected using a Nikon AZ100 macroscope.

**Laser ablation tension measurements.** Embryos were injected at one-cell stage with *mCherry* mRNA and mounted at 80% epiboly for imaging (as described above) on a Zeiss 880 Airyscan 2-photon inverted confocal microscope. An infrared laser tuned to 710–730 nm was used to ablate fluorescently labeled cell interfaces, immediately followed by a quick time-lapse series of ten images with no interval to record recoil after each ablation event. Images were collected using a ×40 water immersion objective lens, and a Zeiss stage heater was used to maintain embryo temperature at 28.5 °C. Approximately 8–12 cell interfaces were ablated per embryo. Image series were analyzed using ImageJ to determine the inter-vertex distance of each cell interface prior to and immediately after ablation and used to calculate recoil distance.

**Calyculin A treatment.** Embryos were mounted in agarose as described above, then embryo medium containing 50 nM Calyculin A (Sigma-Aldrich) was added approximately 30 min prior to the start of imaging and remained throughout the imaging period.

**Image analysis.** ImageJ was used to visualize and manipulate all microscopy datasets. For immunostained embryos, multiple z-planes were projected together to visualize the entire region of interest. For live embryo analysis, a single z-plane through the length of the axial mesoderm was chosen for each time point. When possible, embryo images were analyzed prior to genotyping, and all images were coded during analysis. To measure cell orientation and elongation, the AP axis in all embryo images was aligned prior to manual outlining of cells. A fit ellipse was

used to measure orientation of each cell's major axis and its AR. The TissueAnalyzer ImageJ package[60] was used to automatically segment time-lapse series of axial mesoderm and detect T1 transitions. Boundary straightness was measured by manually tracing the notochord boundary to determine total length, then dividing it by the length of a straight line connecting the ends of the boundary (net length). To assess Pk-GFP localization, isolated cells expressing Pk-GFP were scored according to subcellular localization of GFP signal.

**Quantitative RT-PCR.** Total RNA was isolated from tailbud stage WT and MZ*udu* −/− embryos homogenized in Trizol (Life Technologies), 1 μg of which was used to synthesize cDNA using the iScript kit (BioRad) following manufacturer's protocol. SYBR green (BioRad) qRT-PCR reactions were run in a CFX Connect Real-Time PCR detection system (BioRad) in technical triplicate. Primers used are as follows:

*epcam*: F-TGAGGACGGGGATTGAGAAC
R-GAGCCTGCCATCCTTGTCAT
*itga3b*: F-CCGGTGTTGGGAGAAGAGAC
R-CTTGAAGAAACCACACGAAGGG
*EF1a*: F-CTGGAGGCCAGCTCAAACAT
R-ATCAAGAAGAGTAGTACCGCTAGCATTAC
*Rpl13a*: F-TCTGGAGGACTGTAAGAGGTATGC
R-AGACGCACAATCTTGAGAGCAG

**RNA sequencing and analysis.** RNA for sequencing was isolated from 50 WT or MZ*udu*−/− embryos per sample at tailbud stage according to instructions for the Dynabeads mRNA direct kit (Ambion). Embryos from two clutches per genotype (e.g., WT A and B) were collected, then divided in two (e.g., WT A1, A2, B1, and B2) to yield four independently prepared libraries representing two biological and two technical replicates per genotype. Libraries for were prepared according to instructions for the Epicentre ScriptSeq v2 RNA-seq Library preparation kit (Illumina). Briefly, RNA was enzymatically fragmented prior to cDNA synthesis. cDNA was then 3′ tagged, purified using Agencourt AMPure beads, and PCR amplified, at which time sequencing indexes were added. Indexed libraries were then purified and submitted to the Washington University Genome Technology Access Center for sequencing using an Illumina HiSeq 2500 to obtain single-ended 50 bp reads. Raw reads were mapped to the zebrafish GRCz10 reference genome using STAR (2.4.2a)[61] with default parameters. FeatureCounts (v1.4.6) from the Subread package[62] was used to quantify the number of uniquely mapped reads (phred score ≥10) to gene features based on the Ensembl annotations (v83). Significantly differentially expressed genes were determined by using DESeq2 in the negative binomial distribution model[63] with a cutoff of adjusted *p*-value ≤0.05 and fold change ≥2.0. Heatmaps were built using the heatmaps2 package in R, and other plots were built using the ggplot2 package in R.

**DamID-seq.** *E. coli* DNA adenine methyltransferase (EcoDam) was cloned from the pIND(V5) EcoDam plasmid[21] (a kind gift from Dr. Bas Van Steensel, Netherlands Cancer Institute) by Gibson assembly[64] into a 3′ Gateway entry vector containing 6 Myc tags and an SV40 polyA signal (p3E-MTpA). The resulting p3E-EcoDam-MTpA vector was then Gateway cloned into PCS2 + downstream of *udu* cDNA or *eGFP* to produce C terminal fusions. The resulting *udu-dam-myc* and *gfp-dam-myc* plasmids were linearized by KpnI digestion and transcribed using the mMessage mMachine SP6 in vitro transcription kit (Ambion). WT AB* embryos were injected at one-cell stage with 3 pg mRNA encoding Gon4l-Dam-Myc or with 1 pg encoding GFP-Dam-Myc as controls. The remainder of the protocol was carried out largely as in ref. [21] with modifications. Genomic (g)DNA was collected at tailbud stage using a Qiagen DNeasy kit with the addition of RNAse A. gDNA was digested with DpnI overnight, followed by ligation of DamID adaptors. Unmethylated regions were destroyed by digestion with DpnII, and then methylated regions were amplified using primers complementary to DamID adaptors. Two identical 50 μl PCR reactions were performed and pooled for each sample to reduce amplification bias, and three independent biological replicates were collected per condition. Amplicons were purified using a Qiagen PCR purification kit, then digested with DpnII to remove DamID adaptors. Finally, samples were purified using Agencourt AMPure XP beads and submitted for library preparation and sequencing at the Washington University GTAC using an Illumina HiSeq 2500 to obtain single-ended 50 bp reads.

**Analysis of DamID-seq data.** Raw reads were aligned to zebrafish genome GRCz10 by using bwa mem (v0.7.12) with default parameters[65], then sorted and converted into bam format by using SAMtools (v1.2)[66]. The zebrafish genome was divided into continuous 1000 bp bins, and FeatureCounts (v1.4.6) from the Subread package[62] was used to quantify the number of uniquely mapped reads (phred score ≥10) in each bin. Significantly differentially Gon4l-associated bins were determined by using DESeq2 in the negative binomial distribution model[63] with stringent cutoff: adjusted *p*-value ≤0.01 and fold change ≥4.0. Promoter regions were defined as 2 kb upstream of the transcription start site of a gene based on Ensembl annotations (v83). Wiggle and bigwig files were created from bam files using IGVtools (v2.3.60) (https://software.broadinstitute.org/software/igv/igvtools)

and wigToBigWig (v4)[67], respectively. BigWig tracks were visualized using IGV (v2.3.52)[68].

**Statistical analysis**. Graphpad Prism 6 and 7 software was used to perform statistical analyses and generate graphs of data collected from embryo images. The statistical tests used varied as appropriate for each experiment and are described in the text and figure legends. Data were tested for normal distribution, and non-parametric tests (Mann–Whitney and Kolmogorov–Smirnov) were used for all non-normally distributed data. Normally distributed data with similar variance between groups were analyzed using parametric tests (*T*-tests and ANOVAs). All tests used were two tailed. Differential expression and differential enrichment analysis of RNA and DamID sequencing data were completed as described above. Panther[69] was used to classify differentially expressed genes and to produce pie charts; DAVID Bioinformatic Resources[70] was used for functional annotation analysis.

**Subcloning**. The full-length *udu* open reading frame was subcloned from WT cDNA using following primers:

*udu* cacc F1: CACCATGGGGATGGAAACGCAAGTCTTC
*udu* TGA R: TCAGTCCTGCTCTTCATCAGTGGC
*udu* R: GTCCTGCTCTTCATCAGTGGCCGAC

*udu* cDNAs with or without a stop codon were cloned into the pENTR/D-TOPO (Thermo Fisher) vector, which were then Gateway cloned into PCS2 + upstream of polyA signal or *eGFP* to produce a C terminal fusion.

The full-length *epcam* open reading frame was subcloned from WT cDNA using the following primers:

*epcam* F: GGATCCCATCGATTCGATGAAGGTTTTAGTTGCCTTG
*epcam* R: ACTCGAGAGGCCTTGTTAAGAAATTGTCTCCATCTC

The 5′ portion of the *itga3b* open reading frame was cloned from WT cDNA, and the 3′ portion was cloned from a partial cDNA clone (GE/Dharmacon) using the following primers:

5′ *itga3b* F: TTTGCAGGCGCGCCGGATCCCATCGATTCGATGGCCGGAAAGTCTCTG
5′ *itga3b* R: ATTTGAGTGAGTATGGAATGGAGATGTTGAGCG
3′ *itga3b* F: TCAACATCTCCATTCCATACTCACTCAAATACTCAGG
3′ *itga3b* R: GTTCTAGAGGTTTAAACTCGAGAGGCCTTGTCAGAACTCCTCCGTCAG

The resulting amplicons were Gibson cloned[64] into PJS2 (a derivative of PCS2+) linearized with EcoRI. To create an Itgα3b-GFP fusion, the *itga3b* open reading frame minus the stop codon was amplified from this plasmid and Gibson cloned upstream of an *eGFP* open reading frame into PJS2.

**Data availability**. The authors declare that all data supporting the findings of this study are available within the article and its Supplementary Information files or from the corresponding author upon reasonable request.

Processed RNA-seq and DamID-seq data are available in Supplementary Data files 1-3 of this publication, and raw data have been deposited in the Gene Expression Omnibus (GEO) database under the accession codes: GSE96575 (for RNA-seq) (https://www.ncbi.nlm.nih.gov/geo/query/acc.cgi?token=ipwzmaoqjpajbqd&acc=GSE96575) and GSE96576 (for DamID-seq) (https://www.ncbi.nlm.nih.gov/geo/query/acc.cgi?token=apkloooyvxsjxij&acc=GSE96576).

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

## Acknowledgements

We thank Dr. Bas Van Steensel for EcoDam plasmids, Drs. Christine and Bernard Thisse for WISH probes, Dr. Bo Zhang and Dr. Scott Higdon for bioinformatics help, Dr. Matthew Hass for DamID advice, Bisiayo Fashemi for assistance with image analysis, the Washington University Center for Cellular Imaging for assistance with laser ablation experiments, and the Washington University Genome Technology Access Center for library preparation and sequencing services. The National Institutes of Health grants R01GM55101 and R35GM118179 to L.S.-K. and F32GM113396 to M.L.K.W., and a W.M. Keck Foundation Fellowship to M.L.K.W. in part supported this study.

## Author contributions

M.L.K.W., A.S., and L.S.-K. designed the study. M.L.K.W., A.S., and T.B. performed experiments. C.Y. participated in the initial forward genetic screen. P.G. performed bioinformatic analysis. M.L.K.W. and L.S.-K. wrote the manuscript.

## Additional information

**Competing interests:** The authors declare no competing interests.

