## [Peer Review File(PDF 2005 kb) · Nature Communications]

Reviewers' Comments:

Reviewer #1:

Remarks to the Author:

The manuscript by Solnica-Krezel and coauthors analyzes the function of the chromatin factor *udu/gon4l* in zebrafish gastrulation. The authors provide evidence that *udu* is required for the proper formation of the boundary between notochord and paraxial mesoderm and - partially as a consequence of this - mediolateral (ML) polarization of mesodermal progenitors. They also identify a critical role of integrin α 3b and *epcam* as genomic targets of *udu* in mediating the function of *udu* in boundary formation.

The manuscript is well written and contains a number of interesting observations. While I still have a few suggestions for improvement, overall I think this manuscript would be suitable for publication in Nature Communications.

The proposed model by which *udu* functions in boundary formation and mediolateral cell polarization is quite elaborate and in some places not yet entirely convincing. First, it remains unclear whether *udu* really controls ML cell polarization by its effect on boundary formation. The observation that integrin α 3b can rescue boundary formation but not ML cell polarization questions whether these processes are indeed functionally linked. Likewise, the function of boundary tension in boundary straightness and function remains obscure - are there tension-dependent processes mediated by *epcam* and tension-independent processes mediated by integrin α 3b? Finally, the mechanism(s) by which integrin α 3b 'downstream' of *udu* controls boundary formation is still unclear: is integrin α 3b localizing to this boundary? Would depletion of laminin have similar effects on boundary straightness and function? Some more mechanistic insight into potential effector processes would considerably strengthen this study.

Reviewer #2:

Remarks to the Author:

This manuscript reports the characterisation of a mutant of the chromatin factor *Gon4l* in the zebrafish embryo. The mutant was identified in a screen meant to identify genes enhancing defects produced by weak alleles of PCP mutants. *Gon4l* maternal+zygotic mutant shows shortened axis. Analysis at the cellular level shows defects in convergence extension of the axial mesoderm and delayed/irregular notochord boundary. The function of *Gon4l* appears independent of the PCP pathway. The authors look for direct transcriptional targets of *Gon4l*, and among select *EpCAM* and *Integrin α 3b* as candidates to explain the axis elongation phenotype.

While the basic characterization of the mutant phenotypes is well done, the molecular interpretation of *Gon4l* function as a regulator of axial mesoderm morphogenesis is less convincing, and the potential advances that this study may bring to our knowledge of this developmental process, or of morphogenesis in general, are not apparent.

Transcriptomic / genomic analysis shows that thousands of genes are affected by *Gon4l* mutations. *Gon4l* is highly maternally expressed and that MZ mutants give a stronger phenotype than zygotic mutants. From these two observations alone, it is more than likely that *Gon4l* is a general factor pleiotropically involved a huge number of transcriptional regulations.

The fact that axis elongation is the first apparent process affected, while gastrulation and germ layer determination appear fine may simply be due to global changes in chromatin organization occurring during these stages.

The suggestion that *Gon4l* phenotypes can be explained by deregulation of *EpCAM* and *integrin α 3b* expression is unconvincing and these data would in my view rather add more confusion than improving our knowledge of zebrafish morphogenesis/development.

Specific concerns about the link between *Gon4l* phenotype and the role of candidate targets:

- The data show indeed that *EpCAM* or *integrin α 3b* overexpression perturb the embryo, including

narrowing and elongation of the axis. Yet these are common and not necessarily specific phenotypes, which could be obtained by overexpressing many other components.

- This is here particularly worrying because they are not supported by EpCAM LOF data, since EpCAM depletion fails to rescue loss of Gon4l.
- This should be related to previously published data on zebrafish EpCAM LOF, which failed to detect defects in this process.
- While the authors state that integrin3b depletion rescues notochord boundary straightness, I have worries about the way data on axial mesoderm are here quantified and interpreted. Indeed, the images in fig7h',k' are bizarre, at least on the right sides the boundary shows an empty space. I am sceptical then on using measurements of the straightness of the edge of the notochord as a meaningful readout. By the way, is there any kind of rescue on the general embryo phenotype?

Reviewer #3:

Remarks to the Author:

The manuscript by Williams et al. describes a role for the protein Gon4l in coordinating formation of the notochord boundary with anterior-posterior axis extension in zebrafish. They identified Gon4l as a novel regulator of axis extension via a forward genetic screen for mutants that enhance defects caused by a hypomorphic allele of the knypek/glypican 4 gene. Using germline replacement, the study shows that maternal Gon4l is dispensable for zebrafish development, but that loss of zygotic Gon4l disrupts axial extension during gastrulation (among other phenotypes) without affecting specification of the three germ layers, formation of the embryonic shield, or completion of epiboly. Detailed microscopy studies showed Gon4l is important for notochord boundary formation, specifically affecting the establishment of mediolateral polarity and intercalation by axial mesoderm cells. Additional microscopy studies established that Gon4l and planar cell polarity (PCP) signaling function through independent but partially cooperating pathways, to regulate polarization and intercalation of axial mesoderm cells. Molecular analysis showed gene expression in Gon4l-deficient embryos was widely disrupted, with transcription of some genes being activated, and others repressed. DamID-seq identified genomic binding sites for Gon4l, and comparison of gene expression and DamID-seq data revealed genes that are potentially direct targets for regulation by Gon4l. Among these were the genes encoding EpCAM and Integrin3b, which were both expressed at abnormally high levels due to the loss of Gon4l expression. Given these 2 proteins are regulators of cell adhesion and migration, the authors pursued the notion that increased expression of EpCAM and Integrin3b was in part responsible for the disruption of axial mesoderm cell behavior caused by loss of Gon4l. This was confirmed, to some degree, by genetic studies, particularly in the case of Integrin3b. The final set of data presented suggests that tissue tension at the notochord boundary is reduced by loss of Gon4l, providing additional support for the notion that disruption of mechanisms relying on cell-cell contacts has a role in the impairment of axial extension observed when Gon4l expression is lost.

This is a well-done study that presents data of high quality in a clear and concise manner. The conclusions of the study are for the most part novel and will be of interest to developmental biologists and those interested in the function of Gon4l, which is currently not understood.

Major comments

Some of the data presented in manuscript reproduces phenotypes observed by others (see references 27 and 35), although in those cases the role of maternal Gon4l was not addressed. Nevertheless, it seems appropriate to at least mention the similarities between the results described here and those reported by others.

Admittedly there's not much in the literature regarding Gon4l. However, in addition to zebrafish, Gon4l gene/protein homologs have been studied in plants, worms, flies, and mice. It seems worthwhile for the authors to discuss how their results compare to those obtained in these previously published studies.

Specific comments

In figure 1, panel f, the word "duckling" is misspelled.

For some data, particularly those presented in the supplemental figures, it is unclear how many times experiments were performed or how many embryos were examined.

For figure 5, panel B: it is appreciated that a complex data set is presented and that the graph shown is generated by a computer algorithm. However, it is difficult to determine which group is which in the pie chart because many of the colors used are very similar in tone, at least in a printed version of the figure.

For figure 7, panels c and d: are the differences in *epcam* and *itga3b* expression shown statistically significant?

On page 5, the sentence "all axial mesoderm cells failed to align ML within 31 *kny*^{-/-} embryos at 80% epiboly regardless their position relative to the notochord boundary" is missing the word "of."

We would like to thank the Reviewers for their careful review of our manuscript. We are glad that they considered our work to be novel, of high quality, of interest to the community, and presented in a clear and concise manner. We also appreciate their thoughtful suggestions for how it could be improved. Below, we respond to each Reviewer’s comments point-by-point, including steps we have taken to address their concerns experimentally. We believe these additional data have greatly improved our study, and hope that they answer the questions raised about our original submission to the satisfaction of all three Reviewers.

Reviewer 1

Reviewer 1 has raised a number of interesting experimental questions that are very much in-line with our thinking on this study. The Reviewer wrote:

“First, it remains unclear whether *udu* really controls ML cell polarization by its effect on boundary formation. The observation that integrin alpha3b can rescue boundary formation but not ML cell polarization questions whether these processes are indeed functionally linked.” Likewise, the function of boundary tension in boundary straightness and function remains obscure.”

We have performed a series of experiments to address these concerns. To test the role of notochord boundary tension on boundary straightness and cell polarity, we treated WT and *MZudu*^{-/-} gastrulae with Calyculin A, a Myosin phosphatase inhibitor known to increase myosin contractility and tissue tension in embryos of a variety of species¹⁻³. Although Calyculin A enhanced notochord boundary tension (as measured by recoil distance upon laser ablation) in WT embryos, it did not enhance boundary straightness or ML cell polarity in *MZudu*^{-/-} gastrulae:

These data (presented in our revised Supplemental Figure 8) indicate that increasing tissue tension alone is not sufficient to straighten *MZudu*^{-/-} boundaries. This does not rule out a role for tissue tension in boundary straightness, however, as our new results also show that Calyculin A restored boundary straightness in WT embryos overexpressing *epcam* (discussed further below). This demonstrates that boundary tension and straightness are functionally related, and that additional tension-independent defects likely contribute to boundary phenotypes in *MZudu* mutant gastrulae.

We provided several additional lines of evidence in support of an instructive role for the notochord boundary in ML polarity of axial mesoderm cells. First, at midgastrulation when cells in WT gastrulae exhibit significant mediolateral (ML) alignment, PCP mutant *kny(gpc4)*^{-/-} axial mesoderm cells are nearly randomly oriented. However, at late gastrulation, *kny*^{-/-} axial mesoderm cells located within two cell diameters of the notochord boundary come to exhibit significant ML alignment, whereas more internally located cells remain misaligned (Fig.4h-j). Second, in *MZudu* mutant gastrulae only the cells adjacent to the defective boundary exhibit abnormal ML alignment late in gastrulation (Fig.3d-f). Finally, whereas boundary-adjacent cells become ML polarized in *kny*^{-/-} PCP mutants, they fail to do so in *kny*^{-/-};*udu*^{-/-} compound mutants (Fig.4k-m). Together these data provide strong genetic evidence that the boundary provides some ML cell polarity information.

That partially normalizing boundary straightness or boundary tension in *MZudu* mutants does not suppress ML cell alignment or shape defects (Supplemental Figs.6 & 8) implies that other boundary properties regulated by *Gon4l* are involved, and/or that loss of *Gon4l* affects ML cell alignment and shape in a cell-autonomous manner, e.g. by making cells unable to respond to the boundary-associated polarity cue.

“Are there tension-dependent processes mediated by *epcam* and tension-independent processes mediated by *integrin alpha3b*?”

We appreciate the Reviewer asking this question. Reducing *Itgα3b* but not *EpCAM* in *MZudu* mutants improved boundary straightness (Fig. 7f,h), while overexpression of *epcam* but not *itga3b* reduced boundary tension in WT (Fig. 8c,d,e). This suggests that *EpCAM* and *Itgα3b* affect the boundary via different tension-dependent and -independent mechanisms, respectively. To test this, we treated WT embryos overexpressing either *itga3b* or *epcam* with Calyculin A (to increase tension) and assessed boundary straightness throughout gastrulation:

We found that increasing tension restored boundary straightness in *epcam*, but not *itga3b* overexpressing embryos, indicating that *EpCAM*'s effect on the boundary is tension-dependent while *Itgα3b*'s is tension-independent. These new results (presented in revised Figure 8) are consistent with our laser-ablation tension measurements, and with the hypothesis of distinct molecular mechanisms downstream of these two targets of *Gon4l* regulation.

“The mechanism(s) by which integrin alpha3b 'downstream' of udu controls boundary formation is still unclear: is integrin alpha3b localizing to this boundary?”

To address this question, we created an Integrin α 3b-GFP fusion, which we expressed (by RNA injection) in WT embryos and assessed its localization during gastrulation. Similar fusion proteins have been used to examine not only localization, but also clustering of Integrins in zebrafish embryos⁴. Interestingly, we found that this fusion protein exhibited dynamic localization during gastrulation: it became increasingly localized to the plasma membrane of axial mesoderm cells throughout gastrulation, but was not enriched specifically at the boundary. Furthermore, this relocalization did not occur in presomitic mesoderm or the neural plate, suggesting that Itg α 3b subcellular distribution is regulated in a time- and tissue- dependent manner. These data are included in our revised Figure 7:

“Would depletion of laminin have similar effects on boundary straightness and function?”

To address this important question, we examined whether the effects of Itg α 3b are mediated through interactions with its known ligand, Laminin α 5⁵. We co-injected a *lama5* MO (which we showed to phenocopy the reported *lama5* mutant fin phenotype; Supplemental Fig.6) with excess *itga3b* in WT embryos and measured boundary straightness:

As we observed before, *itga3b* overexpression reduced boundary straightness in WT embryos (Fig.7g,i), but this effect was abrogated by simultaneous loss of Laminin α 5. These data are presented in our revised Figure 7, and we interpret them to mean that the effects of excess Itg α 3b are dependent upon interaction with its ligand. While these results do not reveal the exact mechanism by which excess Integrin disrupts boundary straightness, other cellular contexts may provide clues. Increased expression of Itg α 3 in cancer cells, for example, promotes metastasis and invasion, an effect thought to be mediated by ligand binding⁶. This suggests that

increased Laminin-Integrin interactions may promote epithelial-to-mesenchymal transition and loss of epithelial characteristics. We therefore speculate that a possible mechanism by which excess Itg α 3b reduces boundary straightness may involve suppression of epithelial characteristics in axial mesoderm edge cells.

Together, these results reveal that each of these two identified Gon4l-targets contributes to different aspects of the MZ*udu*^{-/-} boundary phenotype: excess EpCAM disrupts myosin-dependent boundary tension and may contribute to reduced boundary straightness, while excess Itg α 3b impairs boundary straightness through interactions with its laminin ligand in a tension-independent fashion. We anticipate that these experiments will address most of the concerns of Reviewer 1, and believe they have significantly improved our mechanistic understanding of notochord boundary formation and thus the manuscript as a whole.

Reviewer 2

Reviewer 2 expressed some concerns about the pleiotropic nature of the phenotypes we observe in MZ*udu* mutant embryos:

“Transcriptomic / genomic analysis shows that thousands of genes are affected by Gon4l mutations. Gon4l is highly maternally expressed and that MZ mutants give a stronger phenotype than zygotic mutants. From these two observations alone, it is more than likely that Gon4l is a general factor pleiotropically involved a huge number of transcriptional regulations.”

Because the affected gene encodes a ubiquitously expressed chromatin factor, it is not surprising that the resulting phenotypes are numerous and affect a variety of tissues, and we understand the Reviewer’s reservations. However, a better understanding of how cell fate specification and cell movements are coordinated during gastrulation requires analyses at transcriptional and epigenetic levels. Nearly all described loss of function mutations in genes encoding chromatin factors result in pleiotropic phenotypes⁷⁻⁹, clearly demonstrating critical but complex roles for these molecules in regulation of embryogenesis. Yet, as described in our manuscript and further below, we gathered strong evidence that the axis extension defect observed in MZ*udu* mutants during gastrulation is specific and is not the result of a general developmental delay or arrest.

“The fact that axis elongation is the first apparent process affected, while gastrulation and germ layer determination appear fine may simply be due to global changes in chromatin organization occurring during these stages.”

We certainly expect that loss of a chromatin factor would cause global changes in chromatin organization, and that this is very likely responsible for the large-scale gene expression changes we have observed in our mutants. It was the precise goal of this study to identify those changes, and then tie them to specific morphogenetic defects. We have shown that a number of gastrulation processes occur normally in MZ*udu* mutant embryos, including germ layer specification, internalization, and epiboly. Moreover, while extension is impaired in MZ*udu* mutant gastrulae, convergence movements are not affected. Together with our observations that both PCP and Hedgehog signaling are intact, these findings indicate that Gon4l is required for a *specific subset* of cell movements and signaling processes during gastrulation. We have not only identified specific gastrulation defects in MZ*udu* mutants, but also specific downstream genes that contribute to those phenotypes. Moreover, our DamID experiments support the notion that Gon4l directly regulates expression of genes involved in morphogenesis, rather than their altered expression being an indirect effect of global changes in chromatin structure. Therefor, our cell

behavior, RNA-seq and DamID experiments are among the first to provide a link between a chromatin factor and downstream changes in gene expression, and ultimately abnormal gastrulation cell behaviors.

“The suggestion that Gon4l phenotypes can be explained by deregulation of EpCAM and integrin α 3b expression is unconvincing and these data would in my view rather add more confusion than improving our knowledge of zebrafish morphogenesis/development.”

Because our RNA-seq data showed that Gon4l regulates expression of thousands of genes, identifying one or a few genes that could explain all of the many phenotypes exhibited by *MZudu* mutants would be highly unlikely. Nevertheless, we thought it important to use our high-throughput sequencing datasets to make specific hypotheses that could be tested to inform the biology of gastrulation, and sought to identify a molecular link between Gon4l-regulated genes and at least some aspects of the mutant phenotypes. We therefore found it remarkable that modulating the level of one target gene product - Integrin α 3b - could not only phenocopy aspects of *MZudu*^{-/-} defects in WT embryos, but also significantly suppress notochord boundary straightness defects in mutants. We appreciate that this effect was limited to the straightness of the notochord boundary, and did not affect the overall morphology of *MZudu* mutants at later stages; however, this result is important because it provides a molecular basis for a discrete aspect of the complex phenotype. Furthermore, the results of our new experiments probing the role of tissue tension (described above) provide evidence for distinct mechanisms by which *itga3b* and *epcam* each affect notochord boundary formation, demonstrating specific relationships between misregulation of particular target genes and the resulting cellular and tissue phenotypes.

“The data show indeed that EpCAM or integrin α 3b overexpression perturb the embryo, including narrowing and elongation of the axis. Yet these are common and not necessarily specific phenotypes, which could be obtained by overexpressing many other components... This is here particularly worrying because they are not supported by EpCAM LOF data, since EpCAM depletion fails to rescue loss of Gon4l.”

We agree that cell polarity and cell movements in general, and gastrulation movements in particular, are sensitive to variation in gene expression levels. Indeed, one significant insight from our work is that Gon4l is required during gastrulation to prevent precocious, ectopic, and excess gene expression. Therefore, our observations that injection of RNAs encoding EpCAM or Integrin α 3b into WT embryos is sufficient to phenocopy aspects of *MZudu*^{-/-} gastrulation defects is significant because increased *epcam* and *itga3b* transcript levels are observed in *MZudu* mutant gastrulae. The fact that reduction of Integrin α 3b levels suppressed the mutant boundary phenotype provides further support for its specific role in the notochord boundary formation and *MZudu*^{-/-} phenotype.

Whereas reducing EpCAM expression in *MZudu*^{-/-} gastrulae did not suppress the notochord boundary or extension phenotype, in the revised manuscript we present evidence that excess EpCAM is largely causative of a *different* aspect of the *MZudu* mutant phenotype: reduced tension at the boundary. As discussed above, our new data show that increasing tension in *epcam* (but not *itga3b*)-overexpressing WT embryos with Calyculin A rescued their irregular boundaries (Fig.8f-g), implying a tension-dependent role for EpCAM in boundary straightness. Similarly, co-injection of a *lama5* MO rescued the irregular boundaries of *itga3b*-overexpressing embryos (Fig.7i), demonstrating a tension-independent but ligand-dependent role for Itg α 3b at the boundary. Beyond identifying molecular mechanisms downstream of these Gon4l targets, these new experiments further demonstrate that these molecules have distinct activities and can

account for different aspects of the *MZudu* mutant phenotype. Increased tension improved boundary straightness upon overexpression of one but not both of our candidate molecules, indicating a molecular defect specific to excess EpCAM. That the effect of excess *Itga3b* was abrogated by loss of its ligand (*Lama5*) is likewise strong evidence that the phenotypes observed were not simply non-specific effects of RNA overexpression. We therefore propose that these two molecules (*epcam* and *itga3b*) operate via distinct mechanisms (straightness and tension) to regulate proper boundary formation.

We would also like to note that the decreased boundary straightness in *MZudu*^{-/-} gastrulae or embryos overexpressing EpCAM or Integrin α 3b is not a common gastrulation defect. For example, we demonstrate that this defect is not observed in PCP mutant gastrulae (Fig.4).

“[Their data] should be related to previously published data on zebrafish EpCAM LOF, which failed to detect defects in this process.”

We agree that neither C&E nor notochord boundary formation were reported in EpCAM mutant zebrafish¹⁰. However, because *epcam* expression is *increased* in *MZudu* mutants, EpCAM gain-of-function data are much more relevant to our study. Indeed, we have referenced a study in *Xenopus* demonstrating that overexpression of EpCAM disrupts tissue boundary formation¹¹. EpCAM LOF was, however, shown to increase myosin contractility¹², a result that we have referenced and is consistent with our finding that EpCAM overexpression negatively regulates myosin-dependent boundary tension.

Finally:

“While the authors state that integrin α 3b depletion rescues notochord boundary straightness, I have worries about the way data on axial mesoderm are here quantified and interpreted. Indeed, the images in fig7h',k' are bizarre, at least on the right sides the boundary shows an empty space. I am sceptical then on using measurements of the straightness of the edge of the notochord as a meaningful readout.”

We too have noticed these spaces, and have in fact observed them in embryos of many genotypes and conditions, including WT embryos with normal boundary straightness. Hence, these spaces between the axial and paraxial mesoderm do not affect boundary straightness or our ability to quantify it because we always measure along the lateral edge of the axial mesoderm cells, a landmark unchanged by the presence of empty spaces.

Reviewer 3

“Some of the data presented in manuscript reproduces phenotypes observed by others (see references 27 and 35), although in those cases the role of maternal Gon4l was not addressed. Nevertheless, it seems appropriate to at least mention the similarities between the results described here and those reported by others.

Admittedly there's not much in the literature regarding Gon4l. However, in addition to zebrafish, Gon4l gene/protein homologs have been studied in plants, worms, flies, and mice. It seems worthwhile for the authors to discuss how their results compare to those obtained in these previously published studies.”

We agree with Reviewer 3 that few studies in the literature have addressed the function of Gon4l or its homologs, but those that do describe this/these molecules in zebrafish and other organisms

are certainly valuable background for our current study. We acknowledge the important advances made by other researchers, and have therefore elaborated on the results of these previous studies by including references for Gon4l homologs in plants, worms, and flies. We have also added explicit statements in our results sections, when appropriate, indicating that our findings in MZudu mutants are consistent with previously reported results from *udu* zygotic loss-of-function. (Please see page 3 lines 26, 31, 34 and page 4 line 32).

Reviewer 3 also caught a few typos in the manuscript that escaped our attention, and made other helpful suggestions for changes that will improve the readers' understanding of our work. These changes have been made, including re-coloring of pie charts in Figure 5 and addition of sample numbers to both main and supplemental figures.

Additional Note

We have also modified the title of our study to shorten it. The new title is:

Gon4l promotes embryonic axis extension by regulating notochord cell polarity and boundary formation through repression of cell adhesion genes

References

1. Calzolari, S., Terriente, J. & Pujades, C. Cell segregation in the vertebrate hindbrain relies on actomyosin cables located at the interhombomeric boundaries. *EMBO J* **33**, 686-701 (2014).
2. Fernandez-Gonzalez, R., Simoes, S.e.M., Röper, J.C., Eaton, S. & Zallen, J.A. Myosin II dynamics are regulated by tension in intercalating cells. *Dev Cell* **17**, 736-743 (2009).
3. Filas, B.A. *et al.* Regional differences in actomyosin contraction shape the primary vesicles in the embryonic chicken brain. *Phys Biol* **9**, 066007 (2012).
4. Jülich, D., Mould, A.P., Koper, E. & Holley, S.A. Control of extracellular matrix assembly along tissue boundaries via Integrin and Eph/Ephrin signaling. *Development* **136**, 2913-2921 (2009).
5. Carney, T.J. *et al.* Genetic analysis of fin development in zebrafish identifies furin and hemicentin1 as potential novel fraser syndrome disease genes. *PLoS Genet* **6**, e1000907 (2010).
6. Morini, M. *et al.* The alpha 3 beta 1 integrin is associated with mammary carcinoma cell metastasis, invasion, and gelatinase B (MMP-9) activity. *Int J Cancer* **87**, 336-342 (2000).
7. Nambiar, R.M. & Henion, P.D. Sequential antagonism of early and late Wnt-signaling by zebrafish colgate promotes dorsal and anterior fates. *Dev Biol* **267**, 165-180 (2004).
8. Moreno-Ayala, R., Schnabel, D., Salas-Vidal, E. & Lomeli, H. PIAS-like protein Zimp7 is required for the restriction of the zebrafish organizer and mesoderm development. *Dev Biol* **403**, 89-100 (2015).
9. Ma, Y. *et al.* The Chromatin Remodeling Protein Bptf Promotes Posterior Neuroectodermal Fate by Enhancing Smad2-Activated wnt8a Expression. *J Neurosci* **35**, 8493-8506 (2015).
10. Slanchev, K. *et al.* The epithelial cell adhesion molecule EpCAM is required for epithelial morphogenesis and integrity during zebrafish epiboly and skin development. *PLoS Genet* **5**, e1000563 (2009).
11. Maghzal, N., Vogt, E., Reintsch, W., Fraser, J.S. & Fagotto, F. The tumor-associated EpCAM regulates morphogenetic movements through intracellular signaling. *J Cell Biol* **191**, 645-659 (2010).
12. Maghzal, N., Kayali, H.A., Rohani, N., Kajava, A.V. & Fagotto, F. EpCAM controls actomyosin contractility and cell adhesion by direct inhibition of PKC. *Dev Cell* **27**, 263-277 (2013).

Reviewers' Comments:

Reviewer #1:

Remarks to the Author:

The manuscript has been revised along the lines suggested by the different referees. There is, however, still one major point of concern that needs to be addressed: the authors claim that the boundary straightness phenotype in *udu* mutant and *itga3b* overexpression embryos is tension-independent, but fail to show that they actually increase tension by *calA* treatment in those embryos. Perhaps *calA* is not increasing tension in such conditions, and then the conclusion that *udu* and *itga3b* function tension-independently is not correct. The authors need to probe tension in all *calA*-treated conditions in order to conclude about the involvement of tension.

Reviewer #2:

Remarks to the Author:

The revised manuscript has satisfactorily addressed several of the comments of the three reviews. These include in particular supporting evidence for EpCAM overexpression causing notochord boundary defects on actomyosin-dependent tension. Integrin alpha3b overexpression, on the other hand, seems to affect a different, yet uncharacterized process. Altogether, this study presents a large amount of data, the experiments are clean, using state-of-the-art techniques, and the individual results are clean and convincing.

I am afraid, however, that my general conceptual concerns remain unchanged. I do not see any major advance toward understanding of morphogenesis, neither in terms of patterning/cell fate, nor cross-talk between PCP and other pathways, nor at the mechanistic level of axis elongation/boundary formation. On the contrary, as explained below, it seems difficult to put the available data together in a coherent model.

Firstly, to rephrase my previous criticism about the putative role of Gon4l: As shown in Fig.5, this chromatin factor is involved in modulating the expression of hundreds of genes at these early stages. Thus, even if its weak negative effect (<two folds) on EpCAM and integrin alpha3b could explain the axis phenotype, defining Gon4l as a regulator of morphogenesis represents a big leap that I am not ready to do. Alternatively, Gon4l may function as a more general 'modulator', in which case the apparently 'specific' phenotypes may be explained by a higher sensitivity of some processes (here axial mesoderm elongation/boundary straightness) to small changes in gene expression.

Thus what are the evidence for a specific morphogenetic role of Gon4l via EpCAM and and integrin alpha3b? The choice to examine these two targets makes undoubtedly sense, considering that EpCAM overexpression was already known to affect embryonic boundaries (Maghzal et al, 2010) and that *itga3* is a receptor for laminin, which is the most prominent ECM component that accumulates at the notochord boundary. However, at least in the most parsimonious model, one would expect to see these two genes actively and specifically downregulated at the right time and space, and one would predict that Gon4l would specifically regulate these spatial and temporal expressions.

Not only such evidence is missing, but the available data are puzzling, to say the least:

The GFP-integrin alpha3b localization added in the revised manuscript shows increased membrane signal in the axial mesoderm, which is the opposite of what would be expected. Furthermore, this integrin construct is prominently absent from the boundary, which is precisely the region where its substrate laminin is deposited.

EpCAM distribution in the dorsal mesoderm does not seem to be known in fish, but in frogs, EpCAM has been reported to prominently accumulate in the notochord (Maghzal et al, 2013). Assuming a conserved expression of EpCAM between fish and frogs, such enrichment would be hard to reconcile with a negative role in notochord formation.

In terms of downstream mechanisms:

- A role of EpCAM in moderating actomyosin contractility, and a requirement for high contractility is along the notochord boundary have both already been shown in the frog. The present data are

mostly confirmatory in this respect.

- Itga: the GOF and LOF data do convincingly show that itga3b and laminin play a role in notochord boundary straightness, but one is left without a clue of how they would function, especially considering their apparent distinct localization.

In summary, even though the manuscript presents an impressive amount of work, the mechanistic analysis is rather preliminary, and the claim for a specific morphogenetic role for Gon4l remains quite speculative.

Reviewer #3:

Remarks to the Author:

The authors have addressed my concerns regarding the initial version of this manuscript.

Reviewer 1

“The manuscript has been revised along the lines suggested by the different referees. There is, however, still one major point of concern that needs to be addressed: the authors claim that the boundary straightness phenotype in *udu* mutant and *itga3b* overexpression embryos is tension-independent, but fail to show that they actually increase tension by *calA* treatment in those embryos. Perhaps *calA* is not increasing tension in such conditions, and then the conclusion that *udu* and *itga3b* function tension-independently is not correct. The authors need to probe tension in all *calA*-treated conditions in order to conclude about the involvement of tension.”

The Reviewer noted that although we showed that treatment with calyculin A increases tension at the notochord boundary of wild-type embryos, we did not demonstrate that it has a similar effect on either *MZudu* mutants or WT embryos overexpressing *epcam* or *itga3b*. Because this drug has been shown to increase myosin contractility and tissue tension in numerous studies and numerous species¹⁻³ (ours included), we find it unlikely that it would not have this effect within our narrow set of experimental conditions. However, we agree with the Reviewer that this is a formal possibility, and have therefore performed laser cutting tension measurements within WT embryos overexpressing either *epcam* or *itga3b* and treated with calyculin A. We found that, as in control-injected WT embryos, calyculin A increased recoil distance (tension) at the notochord boundary of these embryos compared to untreated *epcam/itga3b*-overexpressing embryos:

These new data are included in Supplementary Fig. 8 and are referenced within the text on page 8, line 36 – page 9 line 2. Because we have now demonstrated that calyculin A increases boundary tension but not straightness in *itga3b*-overexpressing embryos, we are even more confident in our previous interpretation of the data: that excess *itga3b* affects boundary straightness via a tension-independent mechanism.

The Reviewer also requested that we confirm increased boundary tension in *MZudu* mutants treated with calyculin A, but unfortunately we are currently unable to perform these experiments. When germline replacement is used to remove maternal gene function, it is rare for the resulting mosaic animals to become females (as zebrafish with reduced germline tend to develop as males⁴) and even rarer for those females to be productive. In the course of our research described in this manuscript we generated over 20 such WT females with *udu* mutant germline. In recent months, though, despite several rounds of transplantation, we have been unable to generate germ line⁷ replaced females that produce enough embryos to perform the requested experiments. However, the data presented above demonstrate that calyculin A increases boundary tension under a number of conditions, including those designed to phenocopy *MZudu* boundary defects. We hope the Reviewer appreciates the difficulty of these experiments, and is satisfied with the new data we have presented here.

Reviewer 2

“The revised manuscript has satisfactorily addressed several of the comments of the three reviews. These include in particular supporting evidence for EpCAM overexpression causing notochord boundary defects on actomyosin-dependent tension. Integrin alpha3b overexpression, on the other hand, seems to affect a different, yet uncharacterized process. Altogether, this study presents a large amount of data, the experiments are clean, using state-of-the-art techniques, and the individual results are clean and convincing.

I am afraid, however, that my general conceptual concerns remain unchanged. I do not see any major advance toward understanding of morphogenesis, neither in terms of patterning/cell fate, nor cross-talk between PCP and other pathways, nor at the mechanistic level of axis elongation/boundary formation. On the contrary, as explained below, it seems difficult to put the available data together in a coherent model.

Firstly, to rephrase my previous criticism about the putative role of Gon4l: As shown in Fig.5, this chromatin factor is involved in modulating the expression of hundreds of genes at these early stages. Thus, even if its weak negative effect (<two folds) on EpCAM and integrin alpha3b could explain the axis phenotype, defining Gon4l as a regulator of morphogenesis represents a big leap that I am not ready to do. Alternatively, Gon4l may function as a more general ‘modulator’, in which case the apparently ‘specific’ phenotypes may be explained by a higher sensitivity of some processes (here axial mesoderm elongation/boundary straightness) to small changes in gene expression.

Thus what are the evidence for a specific morphogenetic role of Gon4l via EpCAM and and integrin alpha3b? The choice to examine these two targets makes undoubtedly sense, considering that EpCAM overexpression was already known to affect embryonic boundaries (Maghzal et al, 2010) and that itga3 is a receptor for laminin, which is the most prominent ECM component that accumulates at the notochord boundary. However, at least in the most parsimonious model, one would expect to see these two genes actively and specifically downregulated at the right time and space, and one would predict that Gon4l would specifically regulate these spatial and temporal expressions.

Not only such evidence is missing, but the available data are puzzling, to say the least: The GFP-integrin alpha3b localization added in the revised manuscript shows increased membrane signal in the axial mesoderm, which is the opposite of what would be expected. Furthermore, this integrin construct is prominently absent from the boundary, which is precisely the region where its substrate laminin is deposited.

EpCAM distribution in the dorsal mesoderm does not seem to be known in fish, but in frogs, EpCAM has been reported to prominently accumulate in the notochord (Maghzal et al, 2013).

Assuming a conserved expression of EpCAM between fish and frogs, such enrichment would be hard to reconcile with a negative role in notochord formation.

In terms of downstream mechanisms:

- A role of EpCAM in moderating actomyosin contractility, and a requirement for high contractility is along the notochord boundary have both already been shown in the frog. The present data are mostly confirmatory in this respect.

- Itga: the GOF and LOF data do convincingly show that itga3b and laminin play a role in notochord boundary straightness, but one is left without a clue of how they would function, especially considering their apparent distinct localization.

In summary, even though the manuscript presents an impressive amount of work, the mechanistic analysis is rather preliminary, and the claim for a specific morphogenetic role for Gon4l remains quite speculative.”

While we were obviously disappointed that we have not convinced Reviewer 2 of the importance of our study, we were pleased that he/she appreciates the amount of effort we have devoted to it and the quality of the data we have produced.

However, we respectfully disagree with the Reviewer's statement that our work makes no major advance toward understanding of morphogenesis. Our study has characterized a clear role for a chromatin factor during gastrulation morphogenesis, and has identified specific processes that are impaired upon loss of this factor. We do not see the significance of our work in establishing Gon4l as a specific regulator of morphogenesis, but rather in discerning the cellular and molecular mechanisms through which an essential chromatin factor – one that influences expression of numerous genes - regulates morphogenesis during gastrulation. Despite having a broad effect on gene expression during gastrulation, we demonstrate Gon4l regulates subsets of signaling pathways, gastrulation movements, and cell behaviors. Loss of Gon4l leads to reduced axis extension without affecting the other gastrulation movements of internalization, epiboly, or convergence. Our results attribute extension defects in *MZudu* mutants largely to reduced cell intercalation within the axial mesoderm, a phenotype that is likely related to decreased mediolateral (ML) cell orientation. We further found that this polarity defect is especially strong in cells adjacent to the notochord boundary, which is also irregular in *MZudu* mutants. Although a role for the notochord boundary in ML cell polarity has been described in other species^{5,6}, our study provides strong genetic evidence for independent but partially overlapping roles of this boundary-associated cell polarity cue and PCP signaling during convergence & extension. Beyond this assessment of morphogenetic defects, we identified direct and indirect gene targets of Gon4l, and tied two of them directly to the mutant phenotypes we observed. Because notochord boundary defects observed in *MZudu* mutants result from increased expression of both of these Gon4l targets, our study underscores the significance of negative regulation of gene expression by chromatin factors to ensure normal gastrulation movements. This is consistent with numerous additional studies demonstrating the sensitivity of cell movements and cell polarity to both reduced and excess gene function. Together, we feel these genetic, genomic, embryologic, and cell biological studies provide the first example (to our knowledge) of how epigenetic regulation can influence gene expression to affect both embryo patterning and morphogenetic cell behaviors during vertebrate gastrulation.

In regard to the Reviewer's model of the roles of *epcam* and *itga3b* in notochord boundary formation: we understand the logic of the argument that because increased levels of EpCAM and Itg α 3b affect the notochord boundary negatively, they must be negative regulators of boundary formation, and must therefore be lost from the site of boundary formation for development to proceed normally. However, we never considered this as an explanation for our results. Instead, we assume that EpCAM and Itg α 3b are not inherent negative regulators of boundary formation, but can disrupt the process when present in excess. Our DamID experiments indicate that Gon4l likely regulates these genes directly to limit their expression to the proper levels. As noted above, cell polarity and movements during gastrulation are sensitive to both reduced and excess gene expression, including nearly all components of PCP signaling^{7,8} and G α 12/13⁹: too much is just as disruptive as not enough.

Our observation that Itg α 3b-GFP becomes membrane-localized specifically in axial mesoderm cells during the course of gastrulation supports the notion that the subcellular distribution (and presumably activity) of this Integrin is tightly regulated in the axial mesoderm. Because *itga3b* and its ligand *lama5* are expressed within the axial mesoderm of WT embryos^{10,11}, this is exactly where we expected Integrin activity would be evident. We speculate that excess Itg α 3b disrupts boundary formation (by promoting an EMT-like switch, for example¹²) in a way that normal WT levels do not. Whereas Laminin (as the Reviewer notes) is enriched at the nascent notochord boundary, *lama5* transcripts are detected in all notochord cells at the end of gastrulation¹⁰ (and <http://zfin.org/ZDB-PUB-031103-24>). Similarly, our Laminin antibody revealed staining within the notochord in addition to strong accumulation of Laminin at the notochord boundary (Figure 2f,g). Therefore, the interaction between excess Itg α 3b and Lama5 need not necessarily occur at the notochord

boundary, but could affect boundary straightness by interfering with the polarity (or perhaps epithelial character) of all axial mesodermal cells.

We observed that excess EpCAM similarly disrupts boundary development. Indeed, previous studies show that both loss and gain of EpCAM function are deleterious to early development¹³⁻¹⁵. And although EpCAM localizes to the notochord of *Xenopus* tadpoles, as the Reviewer noted, it is actually enriched in the prospective neuroectoderm of early *Xenopus* gastrulae and is largely evenly distributed among the germ layers at neurula stage¹⁴, suggesting that its expression need not be specific to the axial mesoderm to affect its morphogenesis. Moreover, during zebrafish gastrulation *epcam* transcripts are detected almost exclusively in the superficial enveloping layer, but ectopically expressed EpCAM-GFP fusion protein localizes to cell membranes of the superficial and deep gastrula cells¹⁵, consistent with the notion that excess/ectopic EpCAM expression can affect the behavior of notochord cells. Given the well-documented sensitivity of gastrulating embryos to levels (either increased or decreased) of many molecules, we do not find our results to be at all in conflict with the known roles or expression domains of our two candidate molecules.

Reviewer #3

The authors have addressed my concerns regarding the initial version of this manuscript.

References

1. Filas, B.A. *et al.* Regional differences in actomyosin contraction shape the primary vesicles in the embryonic chicken brain. *Phys Biol* **9**, 066007 (2012).
2. Calzolari, S., Terriente, J. & Pujades, C. Cell segregation in the vertebrate hindbrain relies on actomyosin cables located at the interhombomeric boundaries. *EMBO J* **33**, 686-701 (2014).
3. Fernandez-Gonzalez, R., Simoes, S.e.M., Röper, J.C., Eaton, S. & Zallen, J.A. Myosin II dynamics are regulated by tension in intercalating cells. *Dev Cell* **17**, 736-743 (2009).
4. Slanchev, K., Stebler, J., de la Cueva-Méndez, G. & Raz, E. Development without germ cells: the role of the germ line in zebrafish sex differentiation. *Proc Natl Acad Sci U S A* **102**, 4074-4079 (2005).
5. Shih, J. & Keller, R. Patterns of cell motility in the organizer and dorsal mesoderm of *Xenopus laevis*. *Development* **116**, 915-930 (1992).
6. Veeman, M.T. *et al.* Chongmague reveals an essential role for laminin-mediated boundary formation in chordate convergence and extension movements. *Development* **135**, 33-41 (2008).
7. Jessen, J.R. *et al.* Zebrafish trilobite identifies new roles for Strabismus in gastrulation and neuronal movements. *Nat Cell Biol* **4**, 610-615 (2002).
8. Topczewski, J. *et al.* The zebrafish glypican knypek controls cell polarity during gastrulation movements of convergent extension. *Dev Cell* **1**, 251-264 (2001).
9. Lin, F. *et al.* Galpha12/13 regulate epiboly by inhibiting E-cadherin activity and modulating the actin cytoskeleton. *J Cell Biol* **184**, 909-921 (2009).
10. Pollard, S.M. *et al.* Essential and overlapping roles for laminin alpha chains in notochord and blood vessel formation. *Dev Biol* **289**, 64-76 (2006).
11. Thisse, B. & Thisse, C. (ZFIN Direct Data Submission (<http://zfin.org>); 2004).
12. Morini, M. *et al.* The alpha 3 beta 1 integrin is associated with mammary carcinoma cell metastasis, invasion, and gelatinase B (MMP-9) activity. *Int J Cancer* **87**, 336-342 (2000).

13. Maghzal, N., Vogt, E., Reintsch, W., Fraser, J.S. & Fagotto, F. The tumor-associated EpCAM regulates morphogenetic movements through intracellular signaling. *J Cell Biol* **191**, 645-659 (2010).
14. Maghzal, N., Kayali, H.A., Rohani, N., Kajava, A.V. & Fagotto, F. EpCAM controls actomyosin contractility and cell adhesion by direct inhibition of PKC. *Dev Cell* **27**, 263-277 (2013).
15. Slanchev, K. *et al.* The epithelial cell adhesion molecule EpCAM is required for epithelial morphogenesis and integrity during zebrafish epiboly and skin development. *PLoS Genet* **5**, e1000563 (2009).

Reviewers' Comments:

Reviewer #1:

Remarks to the Author:

The authors satisfactorily addressed all my points of criticism, and the manuscript appears now suitable for publication.